# Impact of Copper Nanoparticles on Keratin 19 (KRT19) Gene Expression in Breast Cancer Subtypes: Integrating Experimental and Bioinformatics Approaches

**DOI:** 10.3390/ijms26157269

**Published:** 2025-07-27

**Authors:** Safa Taha, Ameera Sultan, Muna Aljishi, Khaled Greish

**Affiliations:** Princess Al-Jawhara Center for Molecular Medicine, Genetics and Inherited Diseases, Department of Molecular Medicine, College of Medicine and Medical Sciences, Arabian Gulf University, Manama 26671, Bahrain; ameeraa@agu.edu.bh (A.S.); munajma@agu.edu.bh (M.A.); khaledfg@agu.edu.bh (K.G.)

**Keywords:** breast cancer, copper nanoparticles, KRT19, gene expressions, bioinformatics

## Abstract

This study investigates the effects of copper nanoparticles (CuNPs) on KRT19 gene expression in four breast cancer cell lines (MDA-MB-231, MDA-MB-468, MCF7, and T47D), representing triple-negative and luminal subtypes. Using cytotoxicity assays, quantitative RT-PCR, and bioinformatics tools (STRING, g:Profiler), we demonstrate subtype-specific, dose-dependent KRT19 suppression, with epithelial-like cell lines showing greater sensitivity. CuNPs, characterized by dynamic light scattering (DLS) and transmission electron microscopy (TEM) with a mean size of 179 ± 15 nm, exhibited dose-dependent cytotoxicity. Bioinformatics analyses suggest KRT19′s potential as a biomarker for CuNP-based therapies, pending in vivo and clinical validation. These findings highlight CuNPs’ therapeutic potential and the need for further studies to optimize their application in personalized breast cancer treatment.

## 1. Introduction

Copper nanoparticles (CuNPs) have emerged as promising anticancer agents due to their high surface-to-volume ratio, tunable size (179 nm in this study), and ability to induce reactive oxygen species (ROS), leading to oxidative stress and metabolic disruption in cancer cells [1,2]. Synthesized via pulsed electrochemical dissolution, CuNPs exhibit cytotoxicity by impairing cellular structures, including intermediate filaments like Keratin 19 (KRT19), which is overexpressed in aggressive breast cancer subtypes [3]. Despite their potential, the molecular effects of CuNPs, particularly on KRT19 expression, remain underexplored, limiting their development as targeted therapies.

Breast cancer, with approximately 2.3 million new cases in 2022, is a heterogeneous disease encompassing triple-negative (TNBC) and luminal subtypes, each with distinct molecular profiles and therapeutic responses [4,5]. KRT19, a prognostic marker in basal-like and luminal B subtypes, is associated with therapy resistance and metastasis, making it a potential target for novel therapies [6]. While CuNPs’ general cytotoxicity has been documented, their specific impact on KRT19 expression across breast cancer subtypes is unknown, representing a critical research gap.

This study investigates CuNPs’ effects on KRT19 in four breast cancer cell lines (MDA-MB-231, MDA-MB-468, MCF7, and T47D), selected to represent TNBC and luminal subtypes. Using cytotoxicity assays, quantitative RT-PCR, and bioinformatics tools (STRING, g:Profiler), we aim to elucidate subtype-specific responses and evaluate KRT19′s potential as a biomarker for CuNP-based therapies. These findings may inform personalized treatment strategies for breast cancer.

### 1.1. Research Gap

While CuNPs’ anticancer effects have been documented [3], their impact on KRT19 expression in breast cancer remains underexplored. Most studies focus on CuNPs’ general cytotoxicity, with limited investigation into their effects on specific gene expression profiles, particularly KRT19. This gap hinders the development of targeted CuNP-based therapies and the identification of responsive patient subgroups. While this study focuses on breast cancer cell lines to elucidate CuNPs’ effects on KRT19, future investigations will incorporate normal breast epithelial cells to evaluate therapeutic specificity.

### 1.2. Study Rationale and Objectives

This study investigates CuNPs’ effects on KRT19 expression across four breast cancer cell lines (MDA-MB-231, MDA-MB-468, MCF7, and T47D), representing TNBC and luminal subtypes. These cell lines were selected to represent the molecular heterogeneity of breast cancer, with MDA-MB-231 and MDA-MB-468 modeling triple-negative subtypes and MCF7 and T47D representing luminal subtypes, enabling a comprehensive analysis of subtype-specific responses. By integrating cytotoxicity assays, quantitative RT-PCR, and bioinformatics analyses (STRING, g:Profiler), we aim to elucidate subtype-specific responses and evaluate KRT19′s potential as a biomarker for CuNP-based therapies. These findings may advance personalized treatment strategies for breast cancer.

## 2. Results

### 2.1. Characterization of Copper Nanoparticles

CuNPs were synthesized via pulsed electrochemical dissolution, yielding particles with a mean size of 179 ± 15 nm (*n* = 100 particles), as determined by dynamic light scattering (DLS) and confirmed by transmission electron microscopy (TEM). DLS analysis showed a polydispersity index (PDI) of 0.18 ± 0.03, indicating moderate particle uniformity. TEM imaging revealed spherical nanoparticles with a narrow size distribution. These characteristics ensured consistent cellular uptake and cytotoxicity across experiments.

### 2.2. Cytotoxicity and Growth Inhibition by CuNPs

CuNPs’ cytotoxic effects were assessed using the WST-1 assay after 48 h of treatment (5–1800 µg/mL) on the MDA-MB-231, MDA-MB-468 (TNBC), MCF7, and T47D (luminal) cell lines. Dose–response curves (Figure 1) showed dose-dependent inhibition of cell viability (R^2^ = 0.9882–0.9958). IC50 values, calculated via non-linear regression, varied significantly (*p* < 0.001, one-way ANOVA with Tukey’s post hoc test; Table 1). MDA-MB-231 exhibited the highest sensitivity (IC50 = 40.18 µg/mL, 0.632 mM), followed by T47D (112 µg/mL, 1.765 mM), MCF7 (122 µg/mL, 1.920 mM), and MDA-MB-468 (123 µg/mL, 1.936 mM). At 1800 µg/mL, MDA-MB-231 had <10% residual viability, compared with ~30% for MDA-MB-468/MCF7 and ~15% for T47D. Inhibition profiles (Table 1) showed rapid inhibition in MDA-MB-231 (36% ± 3% at 25 µg/mL, >90% at 1800 µg/mL) and delayed but sharp inhibition in T47D (81% ± 2% at 200 µg/mL). These findings indicate subtype-specific responses, with MDA-MB-231′s mesenchymal phenotypes enhancing susceptibility to CuNP-induced oxidative stress.

### 2.3. KRT19 Gene Expression Analysis

KRT19 mRNA expression, assessed by RT-qPCR, showed subtype-specific, dose-dependent suppression (Figure 2). Baseline KRT19 levels (Human Protein Atlas) were highest in T47D (3800 nTPM), followed by MCF7 (2200 nTPM), MDA-MB-468 (2000 nTPM), and MDA-MB-231 (500 nTPM). MDA-MB-468 exhibited the greatest suppression (FC = 0.32 ± 0.05 at IC50, *p* < 0.0001), reaching near-complete suppression at 200 µg/mL (FC ≈ 0). MDA-MB-231 showed consistent reductions (FC = 0.51 ± 0.07 at IC50, *p* < 0.0001), MCF7 a gradual decline (FC = 0.67 ± 0.09 at IC50, *p* < 0.01), and T47D delayed but pronounced suppression (FC = 0.41 ± 0.06 at IC50, *p* < 0.0001). These results suggest that CuNPs preferentially suppress KRT19 in epithelial-like subtypes.

### 2.4. Bioinformatics Analysis

STRING analysis (v12.0) constructed a KRT19 PPI network with 11 high-confidence interactors (confidence score >0.700; Figure 3). The network (11 nodes, 45 edges, PPI enrichment *p*-value = 5.81 × 10^−14^) was clustered into four groups, with Cluster 1 (KRT19, KRT8, KRT18, and EPCAM) linked to cytoskeletal organization and adhesion, Cluster 2 (CEACAM5 and AFP) to tumor progression, Cluster 3 (DSP and ENO2) to cytoskeletal and metabolic processes, and Cluster 4 (C17orf97 and PROM1) to stemness (Table 2). GO enrichment (g:Profiler; Figure 4) identified terms like “intermediate filament cytoskeleton organization” (GO:0045104, *p*-adj = 4.42 × 10^−5^) and “epithelium development” (GO:0060429, *p*-adj = 5.41 × 10^−5^). Reactome pathways included “keratinization” (REAC:R-HSA-6809371, *p*-adj = 2.18 × 10^−5^), and KEGG analysis highlighted “estrogen signaling” (KEGG:04915, *p*-adj = 8.95 × 10^−4^). Co-expression analysis (Figure 5) showed strong correlations with KRT8/KRT18 (cytoskeletal roles) and moderate correlations with EPCAM/LGALS3 (adhesion/progression; Table 3). These findings suggest KRT19′s role in epithelial integrity, disrupted by CuNPs in high-KRT19 subtypes.

## 3. Discussion

This study elucidates CuNPs’ subtype-specific effects on KRT19 expression, revealing a mechanistic link between KRT19 suppression and cytotoxicity. CuNPs significantly suppressed KRT19, with epithelial-like cell lines (MDA-MB-468, fold change: 0.32 at IC50; T47D, 3800 nTPM) showing greater susceptibility than mesenchymal-like MDA-MB-231 (fold change: 0.51, 500 nTPM). Bioinformatics analyses (STRING, g:Profiler) confirmed KRT19′s roles in cytoskeletal organization (GO:0045104) and estrogen signaling (KEGG:04915), suggesting its potential as a biomarker, pending validation. Subtype-specific responses, with initial maintenance in MDA-MB-468 and T47D at low doses, align with KRT19′s prognostic role This differential response positions KRT19 as a potential biomarker for CuNP efficacy in epithelial-like subtypes [7].

Bioinformatics analysis identified KRT19′s role in cytoskeletal organization (GO:0045104, *p*−adj = 4.42 × 10^−5^) and epithelial differentiation (REAC:R-HSA-6805567, *p*−adj = 1.88 × 10^−7^) [8]. KRT19 suppression likely disrupts intermediate filament structures and cell adhesion, enhancing cytotoxicity in high-KRT19-expressing cells (e.g., MDA-MB-468). These findings align with prior research linking KRT19 to epithelial phenotypes and poor prognosis in luminal subtypes [9].

However, KRT19 suppression may confer resistance in mesenchymal cells. Cheung et al. suggested that KRT19 downregulation promotes EMT and chemotherapy resistance in mesenchymal cells like MDA-MB-231, potentially explaining their reduced CuNP susceptibility [10]. Similarly, Ju, J.-H. et al. observed that KRT19-negative cells exhibit increased stemness and apoptosis resistance, which may contribute to MDA-MB-231′s profile [11]. Saha, S.K et al. found that KRT19 silencing enhances metastatic potential via EMT markers like vimentin, further supporting this mechanism [12,13].

The enrichment of the estrogen signaling pathway (KEGG:04915, *p*−adj = 8.95 × 10^−4^) suggests that CuNPs disrupt hormonal regulation in luminal subtypes, amplified by oxidative stress [14]. This aligns with findings that nanoparticle-mediated modulation of estrogen signaling enhances efficacy in hormone-dependent cancers. Keyvani et al. linked KRT19 to estrogen-receptor-positive tumors, supporting this connection [9].

PROM1 (Cluster 4, Figure 3) links KRT19 suppression to cancer stem cell (CSC) dynamics. Zhang et al. found that PROM1-positive cells resist therapy, potentially contributing to MDA-MB-231’s resistance [15]. Conversely, Kawai et al. suggested that KRT19 expression inversely correlates with stemness, implying that CuNP-induced KRT19 suppression in epithelial-like cells (e.g., MDA-MB-468) targets CSCs, enhancing cytotoxicity [16]. Xu et al. supported this, showing that KRT19-positive cells are less stem-like [17].

Clinically, KRT19 could guide CuNP-based therapies, particularly for epithelial-like tumors, potentially combined with endocrine therapies [7]. However, CuNPs’ biodistribution and toxicity require evaluation, as they may accumulate in the liver and spleen [14]. Clinical trials should stratify patients by KRT19 expression and subtype.

### Limitations and Future Directions

The current study provides a robust foundation for understanding the subtype-specific effects of copper nanoparticles (CuNPs) on KRT19 expression in breast cancer cell lines, but its in vitro design presents several limitations. The reliance on in vitro assays (quantitative RT-PCR and WST-1) and bioinformatics analyses (STRING, g:Profiler) limits the relevance of findings to in vivo conditions, potentially overlooking physiological complexities such as biodistribution, systemic metabolism, and tumor microenvironment interactions. Additionally, the use of only four breast cancer cell lines (MDA-MB-231, MDA-MB-468, MCF7, and T47D) may not fully capture the molecular heterogeneity of breast cancer, restricting the generalizability of the results to other subtypes. The single 48 h time point for assessments further constrains insights into the temporal dynamics of CuNPs’ effects on KRT19 expression and cytotoxicity. Moreover, the absence of normal breast epithelial cell controls limited our ability to evaluate the specificity of CuNPs’ effects on cancer cells compared to non-cancerous cells.

To address these limitations, this study represents the first phase (Stage 1) of a multi-stage research project aimed at characterizing CuNPs’ effects on KRT19 expression and establishing its potential as a biomarker for CuNP-based therapies. To further validate and expand upon these findings, Stage 2 of the project will include comprehensive in vivo and biochemical studies. Specifically, we plan to:

Conduct in vivo validation using xenograft models: Xenograft models of the four breast cancer cell lines (MDA-MB-231, MDA-MB-468, MCF7, and T47D) will be established in immunocompromised mice to assess CuNPs’ biodistribution, therapeutic efficacy, and impact on KRT19 expression in a physiological context. KRT19 protein levels will be quantified using immunohistochemistry and Western blotting to confirm the observed mRNA suppression and evaluate its functional consequences in vivo.

Perform mechanistic biochemical analyses: To elucidate the molecular mechanisms underlying CuNPs’ effects, we will conduct biochemical assays, including:

Measurement of reactive oxygen species (ROS) levels to confirm CuNPs’ induction of oxidative stress as a driver of KRT19 suppression.

Analysis of MAPK signaling pathways (e.g., ERK and JNK) using Western blotting and phospho-specific antibodies to investigate downstream signaling effects.

Assessment of epithelial-to-mesenchymal transition (EMT) markers (e.g., vimentin and E-cadherin) to explore the relationship between KRT19 suppression and EMT, particularly in mesenchymal-like cell lines such as MDA-MB-231.

Include normal cell controls: To evaluate the specificity of CuNPs’ effects, normal breast epithelial cell lines (e.g., MCF-10A) will be included in future experiments to compare KRT19 expression and cytotoxicity profiles between cancerous and non-cancerous cells, ensuring that the therapeutic potential of CuNPs is selective for cancerous cells.

Explore clinical and translational applications: Long-term studies will investigate CuNPs’ safety and efficacy in clinical settings through pilot clinical trials, stratifying patients by KRT19 expression and breast cancer subtype. Retrospective analysis of patient samples will validate KRT19′s role as a biomarker for CuNP-based therapies. Additionally, we will explore combination therapies (e.g., with endocrine treatments) and targeted delivery systems to enhance CuNPs’ therapeutic efficacy and minimize off-target effects.

These Stage 2 experiments, which are resource intensive and require additional funding and time, will be reported in a follow-up publication to build on the current findings. The present study establishes a critical foundation by demonstrating CuNPs’ dose-dependent and subtype-specific suppression of KRT19 expression, supported by robust bioinformatic evidence of KRT19′s roles in cytoskeletal organization (GO:0045104, *p*-adj = 4.42 × 10^−5^) and estrogen signaling (KEGG:04915, *p*-adj = 8.95 × 10^−4^). These results provide a hypothesis-generating framework for future mechanistic and translational studies, paving the way for the development of CuNPs as a targeted therapy for breast cancer. By addressing the outlined limitations in Stage 2, we aim to enhance the clinical relevance and our mechanistic understanding of CuNPs’ therapeutic potential in personalized breast cancer treatment.

## 4. Materials and Methods

### 4.1. Cell Culture and CuNP Treatment

Four human breast cancer cell lines (Table 4) (MDA-MB-231, MDA-MB-468 (triple-negative), T47D, and MCF7 (luminal A)) were cultured in Dulbecco’s Modified Eagle Medium (DMEM) (MDA-MB-231, MDA-MB-468) or RPMI-1640 (T47D, MCF7) supplemented with 10% fetal bovine serum (FBS) and 1% penicillin-streptomycin (Thermo Fisher Scientific, Waltham, MA, USA) at 37 °C in 5% CO_2_. CuNPs were prepared in a stock solution and diluted to create concentrations ranging from 1800 µg/mL to 5 µg/mL. Cells were seeded in 96-well plates at 10,000 cells/well to ensure exponential growth during the treatment period. After 24 h of incubation, cells were treated with CuNPs at serial dilutions. Untreated cells served as negative controls. After a 48 h incubation period, cytotoxicity was assessed to determine the half-maximal inhibitory concentration (IC50) for each cell line.

### 4.2. Copper Nanoparticle (CuNP) Synthesis

Copper nanoparticles (CuNPs) were synthesized as described in [1] using a pulsed electrochemical dissolution (PECD) method. Both the cathode and anode were made of copper. An electric current was applied during the precipitation process at 8 volts, with a pulse-on time of 4 ms and a pulse-off time of 8 ms. The reaction was carried out for 30 min with an interelectrode gap of 15 mm. The resulting Cu nanoparticles had an average particle size of 179 nm.

### 4.3. Cytotoxicity and Growth Inhibition Activity: IC_50_ Determination

Cell viability was assessed using the WST-1 assay (Roche Diagnostics, Mannheim, Germany) to determine IC50 values for each treated cell line. Briefly, cells were seeded in 96-well plates at a density of 10,000 cells/well and allowed to adhere for 24 h. After CuNP treatment at various concentrations (1–1800 µg/mL) for 48 h, 10 µL of WST-1 reagent was added to each well, and plates were incubated for 4 h at 37 °C in a humidified 5% CO_2_ atmosphere. Optical density was measured at 450 nm with a reference wavelength of 630 nm using a Synergy HTX plate reader (BioTek, Winooski, VT, USA). Cell viability was calculated as a percentage relative to untreated control cells. IC_50_ values were calculated from dose–response curves using non-linear regression analysis in GraphPad Prism software version 9.0 (GraphPad Software, San Diego, CA, USA). All experiments were performed in triplicate and repeated at least three times independently.

### 4.4. RNA Extraction and cDNA Synthesis

Total RNA was extracted from cells treated with CuNPs for 48 h at different concentrations (MDA-MB-231, 5, 10, 25, 40, and 50 µg/mL, representing below IC_50_, IC_50_, and above IC_50_, respectively; MDA-MB-468, 10, 25, 50, 100, 123, and 200 µg/mL; T47D, 10, 25, 50, 100, 112, and 200 µg/mL; MCF-7, 100, 122, and 200 µg/mL) using the RNeasy Mini Kit (Qiagen, Hilden, Germany) following the manufacturer’s instructions. RNA quality and quantity were assessed using a NanoDrop™ 2000 spectrophotometer (Thermo Fisher Scientific, Waltham, MA, USA), with only samples showing A260/A280 ratios between 1.8 and 2.0 and A260/A230 ratios above 1.7 being used for further analysis. RNA integrity was further confirmed using 1% agarose gel electrophoresis. First-strand cDNA was synthesized from 1 µg of total RNA using the High-Capacity cDNA Reverse Transcription Kit (Applied Biosystems, Waltham, MA, USA). The reverse transcription reaction was performed in a final volume of 20 µL under the following thermal cycling conditions: 25 °C for 10 min (primer annealing), 37 °C for 120 min (reverse transcription), and 85 °C for 5 min (enzyme inactivation). The synthesized cDNA was stored at −20 °C until further use.

### 4.5. Quantitative Real-Time PCR Analysis

Gene expression analysis was performed using PowerUp™ SYBR^®^ Green Master Mix (Applied Biosystems, Foster City, CA, USA) on a StepOnePlus™ Real-Time PCR System. Gene-specific primers for KRT19 and GAPDH (internal control) were designed using NCBI Primer-BLAST and synthesized by Integrated DNA Technologies (IDT, Coralville, IA, USA). The primers were validated for specificity through melt curve analysis and gel electrophoresis. Each PCR reaction (20 µL total volume) contained 2 µL of diluted cDNA (1:5), 10 µL of 2× SYBR^®^ Green Master Mix, 0.5 µM of each forward and reverse primer, and nuclease-free water. The thermal cycling conditions were initial denaturation at 95 °C for 10 min, followed by 40 cycles of denaturation at 95 °C for 15 s and combined annealing and extension at 60 °C for 1 min. A melt curve analysis was performed from 60 °C to 95 °C with 0.3 °C increments. All reactions were performed in technical triplicates with appropriate negative controls. Relative gene expression was calculated using the 2^−ΔΔCt^ method [18] with GAPDH as the reference gene. The PCR efficiency was verified to be between 90 and 110% for all primer pairs (Table 5).

### 4.6. Statistical and Bioinformatics Analysis

Experiments were performed in triplicate, and data are presented as the mean ± standard deviation (SD). Statistical significance was assessed using one-way ANOVA with Tukey’s post hoc test, with *p* < 0.05 considered significant. Graphs and dose–response curves were generated using GraphPad Prism (version 9.0). IC50 values and mean ± SEM were derived from dose–response experiments, with statistical comparisons via one-way ANOVA. STRING network enrichment was assessed with a PPI enrichment *p*-value threshold of *p* < 0.05 [19]. Pathway and functional enrichment analyses used STRING’s Gene Ontology (GO), KEGG, and Reactome databases, with adjusted *p*-values calculated using the Benjamini–Hochberg method. Networks were visualized in Cytoscape (v3.10) with force-directed layouts [20], and pathway enrichment was reconducted via g:Profiler [21].

### 4.7. ΔΔCt Method for Relative Gene Expression

Relative gene expression was calculated using the ΔΔCt method [18]. ΔCt was obtained by subtracting the Ct value of KRT19 from that of GAPDH. ΔΔCt was calculated by subtracting the ΔCt of treated samples from untreated controls. Fold changes were determined as 2^−ΔΔCt^.

## 5. Conclusions

Copper nanoparticles (CuNPs) significantly reduce KRT19 expression in breast cancer cell lines, with greater suppression in epithelial-like subtypes (e.g., MDA-MB-468, FC = 0.32 at IC50) compared with mesenchymal-like subtypes (e.g., MDA-MB-231, FC = 0.51). KRT19, an intermediate filament protein, contributes to cytoskeletal organization and epithelial integrity in luminal subtypes (e.g., MCF7 and T47D) and is associated with cell adhesion and tumor progression via interactions with KRT8, KRT18, and EPCAM, as suggested by STRING analysis. Bioinformatics analyses further indicate KRT19′s involvement in cytoskeletal organization (GO:0045104, *p*-adj = 4.42 × 10^−5^) and estrogen signaling (KEGG:04915, *p*-adj = 8.95 × 10^−4^), supporting its potential as a biomarker for CuNP-based therapies. These findings, derived from RT-qPCR, WST-1 assays, and bioinformatics tools, are preliminary and require in vivo and clinical validation in Stage 2 of this project. This study provides a hypothesis-generating framework for developing CuNPs as targeted therapies for breast cancer, particularly in epithelial-like subtypes.

## Figures and Tables

**Figure 1 ijms-26-07269-f001:**
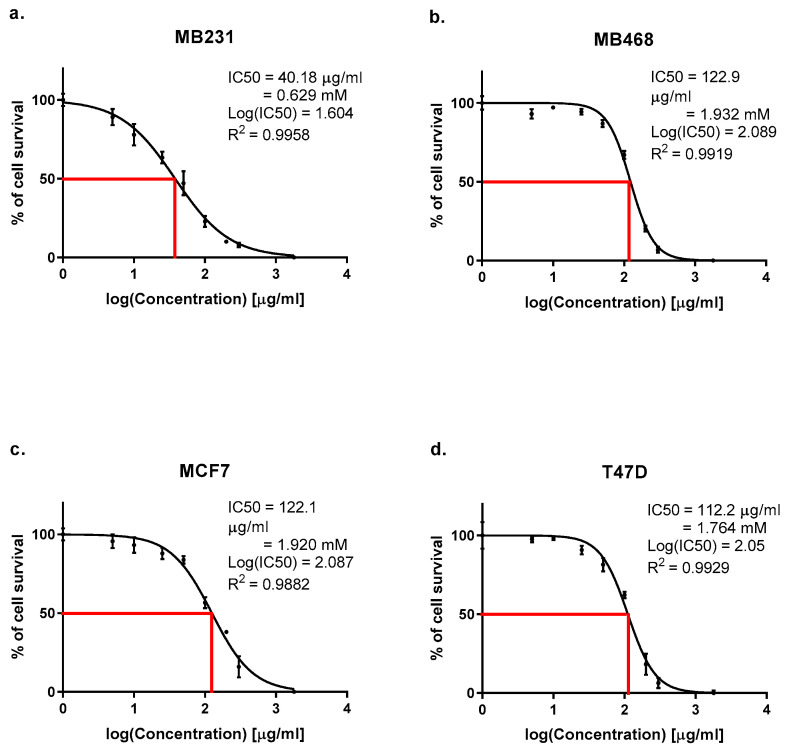
Dose–response curves illustrating the effect of copper nanoparticles (CuNPs) at concentrations ranging from 5 to 1800 µg/mL over 48 h on cell viability in the (**a**) MDA-MB-231, (**b**) MDA-MB-468, (**c**) MCF7, and (**d**) T47D breast cancer cell lines, assessed by the WST-1 assay. Data represents the standard deviation (SD) from three independent experiments (*n* = 3). IC50 values were calculated using non-linear regression in GraphPad Prism (version 8.0). Statistical significance was determined by one-way ANOVA with Tukey’s post hoc test (*p* < 0.001).

**Figure 2 ijms-26-07269-f002:**
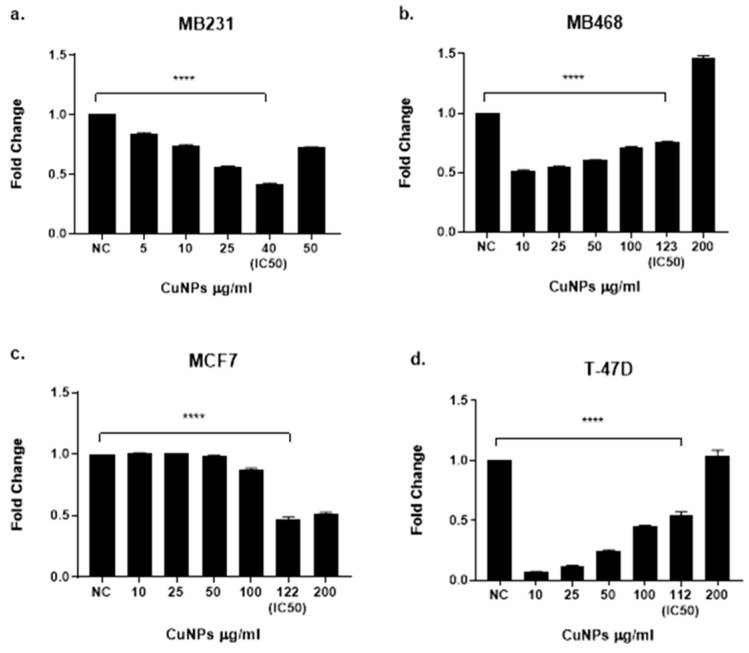
Subtype-specific modulation of KRT19 mRNA expression. Dose–response curves showing fold changes in KRT19 mRNA levels relative to untreated controls in (**a**) MDA-MB-231, (**b**) MDA-MB-468, (**c**) MCF7, and (**d**) T47D cells treated with CuNPs (5–1800 µg/mL, 48 h). IC50 values are indicated by vertical dashed lines. Data represents the mean ± SD from three independent experiments (*n* = 3 technical replicates). Statistical significance was determined by one-way ANOVA with Tukey’s post hoc test (**** *p* < 0.0001 vs. control).

**Figure 3 ijms-26-07269-f003:**
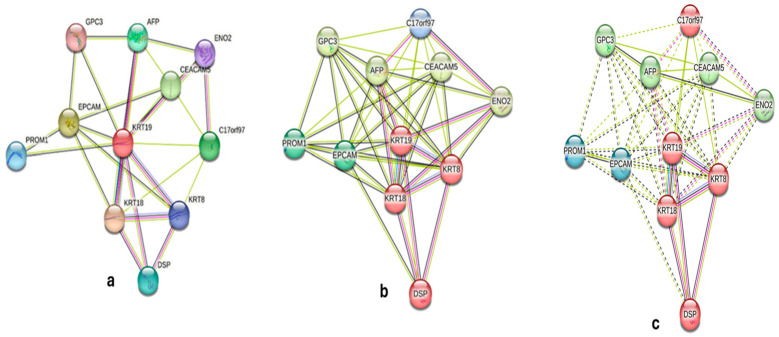
STRING PPI network of KRT19 interactors. (**a**) Network with k-means clustering into four color-coded clusters, showing KRT19 and 10 interactors (confidence score > 0.700). (**b**,**c**) Network with edge thickness proportional to interaction confidence. Nodes represent proteins (KRT19, KRT8, KRT18, EPCAM, CEACAM5, C17orf97, AFP, DSP, PROM1, ENO2, and GPC3); edges indicate interactions. Data sourced from STRING (v12.0).

**Figure 4 ijms-26-07269-f004:**
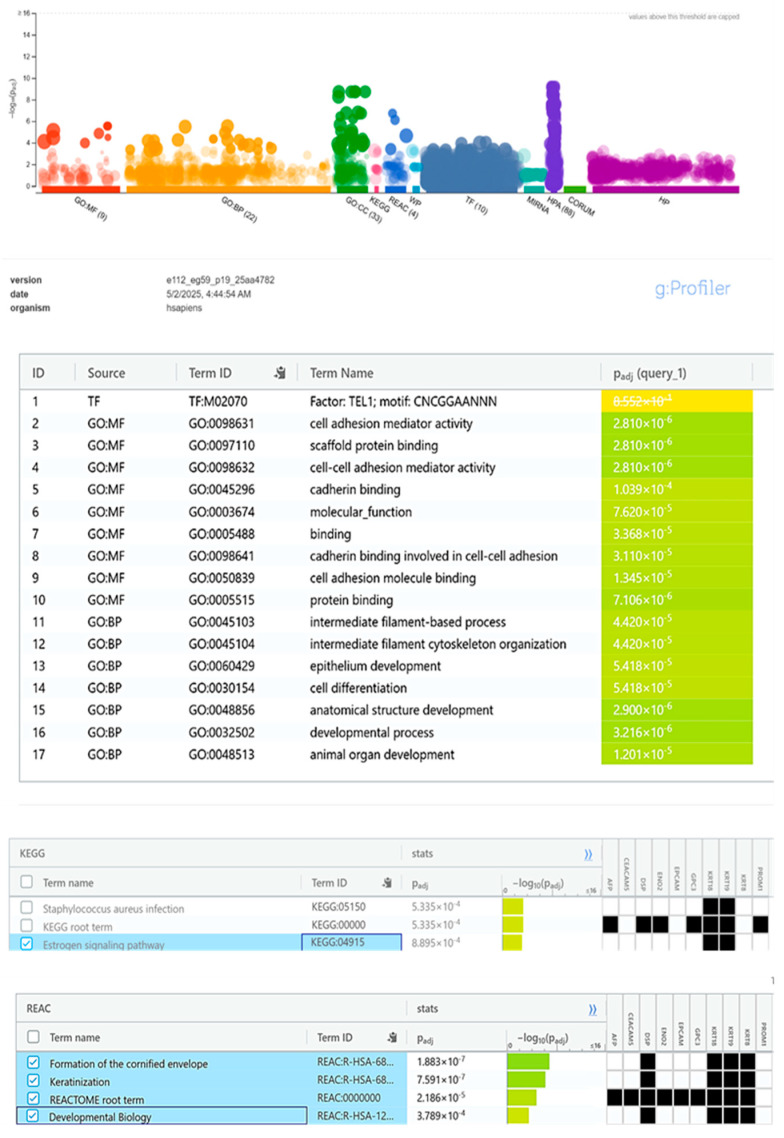
Comprehensive enrichment analysis of KRT19 interactors. Visualization of enriched GO molecular function, biological process, KEGG, and reactome terms using g:Profiler. Circle size and color denote gene count and FDR, respectively.

**Figure 5 ijms-26-07269-f005:**
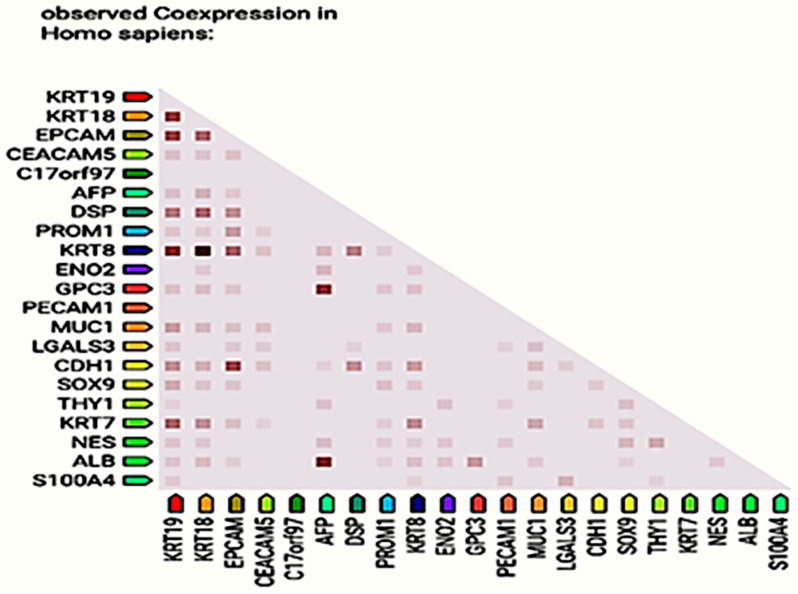
KRT19 co-expression heatmap. The heatmap illustrates the co-expression strength of KRT19 with other genes (dark red, strong; medium pink, moderate; light pink/white, weak). Data sourced from STRING (v12.0) and ProteomeHD.

**Table 1 ijms-26-07269-t001:** Cytotoxicity, inhibition profiles, and IC50 values for breast cancer cell lines treated with CuNPs.

Cell Line	Hormone Receptors	Subtype	IC50 (µg/mL)	IC50 (mM)	Viability at 1800 µg/mL (%)	Sensitivity Rank	R^2^
MDA-MB-231	ER−, PR−, HER2−	TNBC	40	0.629	<10	Highest (1)	0.9958
MDA-MB-468	ER−, PR−, HER2−	TNBC Basal	123	1.932	~30	Low (4)	0.9919
MCF7	ER+, PR+, HER2−	Luminal	122	1.920	~28	Low (3)	0.9882
T47D	ER+, PR+, HER2−	Luminal	112	1.764	~15	Moderate (2)	0.9929

**Table 2 ijms-26-07269-t002:** KRT19 protein–protein interaction network interactors.

Gene	Confidence Score	Role in Breast Cancer
KRT8	0.95	Cytoskeletal organization
KRT18	0.92	Cytoskeletal organization
EPCAM	0.888	Cell adhesion, epithelial marker
CEACAM5	0.85	Tumor progression
C17orf97	0.8	Cytoskeletal organization
AFP	0.78	Tumor marker
DSP	0.77	Cell adhesion
PROM1	0.76	Cancer stem cell marker
ENO2	0.75	Tumor progression
GPC3	0.74	Tumor progression

**Table 3 ijms-26-07269-t003:** Genes co-expressed with KRT19.

Gene	Co-Expression Strength	Role in Breast Cancer
KRT18	Strong (dark red)	Cytoskeletal organization
KRT8	Strong (dark red)	Cytoskeletal organization
EPCAM	Moderate (medium pink)	Cell adhesion, epithelial marker
LGALS3	Moderate (medium pink)	Metastasis, cell migration

**Table 4 ijms-26-07269-t004:** Breast cancer cell line characteristics.

Characteristic	MDA-MB-231	MDA-MB-468	T47D	MCF7
ATCC Catalog	HTB-26	HTB-132	HTB-133	HTB-22
Molecular Subtype	Triple-negative	Triple-negative (Basal A)	Luminal	Luminal
Hormone Receptors				
ER	Negative	Negative	Positive	Positive
PR	Negative	Negative	Positive	Positive
HER2	Negative	Negative	Negative	Negative
Key Markers				
EGFR	Positive	Strongly Positive	Low	Low
Vimentin	Positive	Less mesenchymal	-	Low
CD44	Positive	Lower expression	-	-
Ki67	Low	Moderate	Moderate	Moderate
CK5/6	-	Positive	Low	Low
EpCAM	-	Positive	Positive	Strongly Positive
KRT19	Low	Moderate	Positive	Strongly Positive
Special Features				
Phenotype	CD44+/CD24−/low	Grape-like morphology	Epithelial	Drug resistant (ABCG2+)
Other Features	Mesenchymal, Invasive	Lacks E-cadherin, Wild-type p53	Differentiated epithelial	Differentiated epithelial

**Table 5 ijms-26-07269-t005:** Primer sequences.

Gene	Forward Primer (5′-3′)	Reverse Primer (5′-3′)	Amplicon Size (bp)	Tm (°C)
KRT19	GCGAGCTAGAGGTGAAGATC	AGTGCTCCCAGACGCAAG	198	60
GAPDH	GAAGGTGAAGGTCGGAGTC	GAAGATGGTGATGGGATTTC	226	60

## Data Availability

Available upon request (Corresponding author).

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
