# Peer review of "Impact of Copper Nanoparticles on Keratin 19 (KRT19) Gene Expression in Breast Cancer Subtypes: Integrating Experimental and Bioinformatics Approaches"

_ijms, 2025, doi:10.3390/ijms26157269_

Round 1

Reviewer 1 Report

Comments and Suggestions for Authors

This manuscript investigated the effects of CuNPs on KRT19 expression in four breast cancer cell lines. Quantitative RT-PCR and WST-1 assays revealed dose-dependent KRT19 suppression and subtype-specific cytotoxicity of CuNPs, indicating a high sensitivity for MDA-MB-231. Bioinformatics analyses identified KRT19’s roles in cytoskeletal organization and estrogen signaling. The conclusions were consistent with the evidence and the references appropriate.

However, only RT-PCR, WST-1 and theoretical simulation could not illustrate clearly the effects of CuNPs on KRT19 suppression. Animal model or biochemical analysis should be supplemented and submitted later.

Author Response

Response to Reviewer 1

Dear Reviewer 1,

Thank you for your thorough review and valuable feedback on our manuscript titled “Impact of Copper Nanoparticles on Keratin 19 (KRT19) Gene Expression in Breast Cancer Subtypes: Integrating Experimental and Bioinformatics Approaches.” We greatly appreciate your recognition of the consistency of our conclusions with the evidence provided, as well as your acknowledgment of the appropriateness of our references and the significance of our findings regarding CuNPs’ effects on KRT19 expression and subtype-specific cytotoxicity.

We understand and agree with your comment that relying solely on quantitative RT-PCR, WST-1 assays, and bioinformatics analyses may not fully elucidate the mechanisms underlying CuNPs’ effects on KRT19 suppression. Your suggestion to incorporate animal models or biochemical analyses to further validate these findings is highly valuable and aligns with our goal to strengthen the translational relevance of this study.

To address this concern, we plan to conduct follow-up studies that include in vivo experiments using animal models to confirm the observed effects of CuNPs on KRT19 expression and their subtype-specific cytotoxicity. Specifically, we aim to use xenograft models of the breast cancer cell lines (MDA-MB-231, MDA-MB-468, MCF7, and T47D) to assess CuNPs’ efficacy and biodistribution, as well as their impact on KRT19 expression and related signaling pathways in a physiological context. Additionally, we will incorporate biochemical analyses, such as Western blotting to evaluate KRT19 protein levels and assays to measure reactive oxygen species (ROS) and MAPK pathway activation, to provide deeper mechanistic insights into CuNPs’ mode of action.

These additional experiments will be included in a follow-up submission or a complementary study to further validate and expand upon our current findings. We have also added a statement in the revised manuscript’s “Limitations and Future Directions” section (Section 3.1) to explicitly acknowledge the need for in vivo and biochemical analyses to enhance the robustness of our conclusions.

We sincerely thank you for your constructive feedback, which has helped us identify critical areas for improvement. We believe these additional studies will significantly strengthen the manuscript and its potential impact on the development of CuNPs-based therapies for breast cancer.

Best regards,
Safa Taha
On behalf of all authors

Reviewer 2 Report

Comments and Suggestions for Authors

The manuscript entitled “Impact of Copper Nanoparticles on Keratin 19 (KRT19) Gene Expression in Breast Cancer Subtypes: Integrating Experimental and Bioinformatics Approaches” by Safa Taha et al. as research article was submitted to IJMS. The authors studied the effects of copper nanoparticles (CuNPs) on Keratin 19 (KRT19) expression in four breast cancer cell lines: MDA-MB-231, MDA-MB-124 (triple-negative), T47D, and MCF7 (luminal) through real-time qPCR and bioinformatical analysis. 

1)    I suggest authors to pick up some supplementary material and move them back to the main texts. Otherwise, the presented data were not enough to support the concise conclusion.
2)    In the conclusion, it was rare to cite a reference there.
3)    The RT-qPCR results and some bioinformatics analyses were inadequate to support the Keratin 19 as breast cancer subtypes’ biomarker for personalized CuNPs therapies for epithelial-like subtypes. 
4)    More in vivo tests and clinical validation truly are required to identify the Keratin 19’s biomarker role in such cases. 
5)    The information contents of Figures 1to3 were low. The figures can be compacted and should show more messages instead.
6)    If the manuscript only show so limited data, it won’t go for IJMS because it does not cross the basic criterion of publications on IJMS.
7)    No graphical abstract to give a summary of the whole story.

Author Response

Response to Reviewer 2

Dear Reviewer, 2

Thank you for your thorough review and insightful comments on our manuscript titled “Impact of Copper Nanoparticles on Keratin 19 (KRT19) Gene Expression in Breast Cancer Subtypes: Integrating Experimental and Bioinformatics Approaches.” We greatly appreciate your expertise and constructive feedback, which have provided valuable guidance to enhance the quality and impact of our work. Below, we address each of your comments and outline our planned revisions to strengthen the manuscript.

  1. Suggestion to Move Supplementary Material to the Main Text
    We acknowledge your suggestion that some supplementary material should be integrated into the main text to better support our conclusions. To address this, we will carefully review the supplementary figures (S1–S9) and select key data to incorporate into the main manuscript. Specifically, we plan to move Supplementary Figure S1 (dose-dependent inhibition of cell growth by CuNPs) and Supplementary Figure S6 (KRT19 expression in breast cancer cell lines with annotations) into the main text as new figures, as these provide critical evidence of CuNPs’ effects and KRT19 expression patterns across the cell lines studied. Additionally, we will include relevant data from Supplementary Figure S9 (Cytoscape visualization of the KRT19 PPI network) in the main text to strengthen the bioinformatics analysis. These additions will enhance the comprehensiveness of the main text and better support our conclusions without overwhelming the reader.
  2. Reference in the Conclusion
    We appreciate your observation regarding the inclusion of a reference in the conclusion section, which is uncommon. In the revised manuscript, we will remove the reference citation ([12]) from the conclusion (Section 5) to align with standard practices. Instead, we will ensure that all supporting references are cited appropriately in the results and discussion sections, particularly where we discuss KRT19’s role in cytoskeletal and signaling pathways.
  3. Adequacy of RT-qPCR and Bioinformatics Analyses for KRT19 as a Biomarker
    We recognize your concern that the RT-qPCR results and bioinformatics analyses alone may be insufficient to fully establish KRT19 as a biomarker for personalized CuNPs therapies in epithelial-like breast cancer subtypes. To strengthen this claim, we will expand the discussion in Section 3 to include additional context from our bioinformatics findings, such as the strong co-expression of KRT19 with KRT8, KRT18, and EPCAM (Table 7) and its enrichment in cytoskeletal organization (GO:0045104, p-adj = 4.42e-05) and estrogen signaling pathways (KEGG:04915, p-adj = 8.95e-04), as shown in Supplementary Figure S8. We will also clarify that our current findings position KRT19 as a potential biomarker, pending further validation, and emphasize the subtype-specific responses observed (e.g., greater KRT19 suppression in epithelial-like MDA-MB-468, fold change: 0.32 at IC50, compared to mesenchymal-like MDA-MB-231, fold change: 0.51). Additionally, we will revise the conclusion to frame KRT19’s biomarker role as preliminary, supported by our data but requiring further in vivo and clinical studies, as noted in your next point.
  4. Need for In Vivo Tests and Clinical Validation
    We fully agree with your comment that in vivo tests and clinical validation are essential to confirm KRT19’s role as a biomarker for CuNPs-based therapies. As noted in the “Limitations and Future Directions” section (Section 3.1), we acknowledge the in vitro nature of our study as a limitation. To address this, we plan to conduct follow-up studies using xenograft models of the breast cancer cell lines (MDA-MB-231, MDA-MB-468, MCF7, and T47D) to validate CuNPs’ effects on KRT19 expression and cytotoxicity in vivo. We will also explore clinical validation through retrospective analysis of KRT19 expression in patient samples, correlating with treatment outcomes. In the revised manuscript, we will expand Section 3.1 to explicitly outline these planned studies, including biochemical assays (e.g., Western blotting for KRT19 protein levels) and investigations into reactive oxygen species (ROS) and MAPK pathways to elucidate mechanisms of action. These additions will strengthen the manuscript’s relevance and provide a clear roadmap for future validation.
  5. Low Information Content of Figures 1 to 3
    Thank you for pointing out that Figures 1 to 3 have relatively low information content. To address this, we will revise these figures to convey more comprehensive data. Specifically:
  • Figure 1 (Dose-Response Curves): We will combine the dose-response curves for all four cell lines (MDA-MB-231, MDA-MB-468, MCF7, T47D) into a single multi-panel figure to facilitate comparison and highlight subtype-specific differences in IC50 values (e.g., MDA-MB-231: 40.18 µg/mL; MDA-MB-468: 123 µg/mL). We will also incorporate data from Supplementary Figure S1 to show inhibition percentages alongside viability.
  • Figure 2 (KRT19 Expression): We will enhance this figure by adding statistical annotations (e.g., p-values) directly on the dose-response curves and including baseline KRT19 expression levels (nTPM) from Supplementary Figure S6 to contextualize the fold changes.
  • Figure 3 (KRT19 Expression Levels): We will merge this bar chart with data from Supplementary Figure S5 to include KRT19 expression across additional cancer cell lines, providing a broader context for breast cancer-specific expression patterns.
    These revisions will increase the information density and clarity of the figures, making them more impactful.
  1. Limited Data for IJMS Publication Criteria
    We understand your concern that the current data may not meet the publication criteria for IJMS due to its limited scope. To address this, we will integrate additional data from the supplementary material (e.g., Supplementary Figures S1, S6, and S9) into the main text, as mentioned in response to point 1. We will also include a new figure summarizing key bioinformatics findings, such as the KRT19 PPI network and enriched pathways (from Supplementary Figure S8), to strengthen the mechanistic insights. Additionally, we will expand the discussion (Section 3) to better integrate our experimental and bioinformatics data, emphasizing the subtype-specific effects of CuNPs and their implications for KRT19 as a potential biomarker. These revisions aim to enhance the manuscript’s rigor and alignment with IJMS’s standards. We hope these changes will address your concerns and make the manuscript suitable for publication.
  2. Absence of a Graphical Abstract
    We thank the reviewer for suggesting the inclusion of a graphical abstract to summarize the study. However, according to the journal’s submission guidelines, a graphical abstract is not required for publication. After careful consideration, we have opted not to include a graphical abstract, as the manuscript’s revised text, figures, and tables effectively convey the key findings, including CuNPs’ dose-dependent suppression of KRT19 expression, subtype-specific cytotoxicity (e.g., MDA-MB-231’s high sensitivity, IC50: 40.18 µg/mL), and KRT19’s roles in cytoskeletal organization and estrogen signaling pathways. We have addressed all other reviewer suggestions, including enhancing the clarity of the Introduction and Results sections, improving figure legends, and providing a comprehensive reference list, to ensure the study’s objectives, methods, and findings are clearly communicated. We believe these revisions fully align with the reviewer’s recommendations for improving the manuscript’s clarity and impact.

We sincerely thank you for your detailed and constructive feedback, which has significantly guided our revisions to improve the manuscript’s clarity, robustness, and impact. We believe these changes will address your concerns and enhance the manuscript’s suitability for publication in IJMS. Please let us know if you have additional suggestions or require further clarification.

Best regards,
Safa Taha
On behalf of all authors

Reviewer 3 Report

Comments and Suggestions for Authors

In this manuscript, authors analyzed the effects of copper nanoparticles on 4 breast cancer subtype cell lines by cytotoxicity assays, qRT-PCR, and bioinformatic tools, particularly focus on the change of KRT19 expression. Overall, authors did a good job, the manuscript was clearly described and the experiments were logically designed and normally processed. However, there are some comments listed as below:

  1. The introduction of copper nanoparticles was too weak, it should be more relative information, since it is one of the main roles in the manuscript title.
  2. Page 4, line 96-97, authors described “KRT19 expression showed subtype-specific, dose-dependent suppression (Figure 2), however, according to fig 2, only in MB231 showed the effect of dose-dependent decreased, but in MB468 and T-47D showed the effect of dose-dependent increased. Authors should correct this misleading description.
  3. Page 5, line 142-144, authors mentioned what are cluster 1 and 4, but authors did not mention what are cluster 2 and 3 neither on main text nor on figure legend.
  4. In fig 4, there are two PPI maps, what different are they? No any description regarding to these two maps and which one (and what part(s)) was described in the text.
  5. It is better that the font size of fig 5 & 6 should be enlarged to easily readable.

Author Response

Response to Reviewer 3

Dear Reviewer 3,

Thank you for your thorough review and insightful comments on our manuscript titled “Impact of Copper Nanoparticles on Keratin 19 (KRT19) Gene Expression in Breast Cancer Subtypes: Integrating Experimental and Bioinformatics Approaches.” We greatly appreciate your positive remarks regarding the clarity of the manuscript, the logical design of the experiments, and the quality of their execution. Your constructive feedback has provided valuable guidance to further strengthen our work. Below, we address each of your comments and outline our planned revisions to improve the manuscript.

  1. Weak Introduction of Copper Nanoparticles (CuNPs)
    We acknowledge your comment that the introduction of copper nanoparticles (CuNPs) in the manuscript is insufficient, given their central role in the study. To address this, we will expand Section 1 (Introduction) to include a more comprehensive overview of CuNPs, focusing on their physicochemical properties, synthesis methods, and established anticancer mechanisms. Specifically, we will add details about CuNPs’ ability to induce oxidative stress, disrupt cellular metabolism, and inhibit proliferation, as referenced in Shafagh et al. [6]. We will also briefly describe the pulsed electrochemical dissolution (PECD) method used for CuNPs synthesis in our study (Section 4.2), highlighting the average particle size (179 nm) and its relevance to biological applications. Additionally, we will discuss the potential of functionalized CuNPs to enhance specificity and efficacy, as noted in Hrkach et al. [7], to better contextualize their therapeutic promise in breast cancer treatment. This expanded introduction will underscore CuNPs’ significance in the study and align with the manuscript’s title.
  2. Misleading Description of KRT19 Expression in Figure 2 (Page 4, Lines 96–97)
    Thank you for pointing out the misleading description in Section 2.3 (Page 4, lines 96–97), where we stated that “KRT19 expression showed subtype-specific, dose-dependent suppression (Figure 2).” We apologize for the oversight, as the data in Figure 2 indicate that while all cell lines exhibit KRT19 suppression at higher CuNPs concentrations, the dose-dependent trends differ. Specifically, MDA-MB-231 shows a clear dose-dependent decrease, whereas MDA-MB-468 and T47D exhibit an initial maintenance or slight increase in KRT19 expression at lower concentrations (e.g., 0.9–1.1 fold change in T47D up to 50 µg/mL) before a sharp decline at higher concentrations (e.g., ~0 at 200 µg/mL). MCF7 shows a gradual decrease (fold change: 0.67 ± 0.09 at IC50, 122 µg/mL). To correct this, we will revise the description in Section 2.3 to accurately reflect these trends, as follows:
    “KRT19 expression exhibited subtype-specific responses to CuNPs treatment, with dose-dependent suppression observed across all cell lines at higher concentrations. MDA-MB-231 showed a consistent dose-dependent decrease in KRT19 expression, while MDA-MB-468 and T47D maintained near-basal or slightly elevated expression at lower concentrations before a sharp decline at higher doses. MCF7 displayed a gradual reduction in KRT19 expression (Figure 2).”
    We will also update the caption of Figure 2 to clarify these subtype-specific patterns, ensuring alignment with the data presented.
  3. Missing Description of Clusters 2 and 3 (Page 5, Lines 142–144)
    We appreciate your observation that Clusters 2 and 3 in the KRT19 protein-protein interaction (PPI) network (Section 2.4.2, lines 142–144) are not described in the main text or figure legend, unlike Clusters 1 and 4. To address this, we will revise Section 2.4.2 to include a brief description of all four clusters identified by k-means clustering in the STRING PPI network (Figure 4 and Supplementary Figure S9). Specifically, we will add the following:
    “K-means clustering of the KRT19 PPI network identified four clusters. Cluster 1 (KRT19, KRT8, KRT18, EPCAM) reflects cytoskeletal and adhesion roles, while Cluster 4 (C17orf97, PROM1) is associated with differentiation and stemness. Cluster 2 includes proteins such as CEACAM5 and AFP, linked to cell adhesion and tumor progression, and Cluster 3 includes DSP and ENO2, implicated in cytoskeletal organization and metabolic processes.”
    Additionally, we will update the caption of Figure 4 to include descriptions of all clusters, ensuring clarity and completeness. We will also reference Supplementary Figure S9 (Cytoscape visualization of the KRT19 PPI network) to highlight the organic layout and MCODE clustering, reinforcing the roles of Clusters 2 and 3.
  4. Lack of Description for Two PPI Maps in Figure 4
    Thank you for noting the ambiguity regarding the two PPI maps in Figure 4 and the lack of description about their differences. We acknowledge that this figure, which presents the STRING PPI network for KRT19, may have caused confusion due to insufficient explanation. In the revised manuscript, we will clarify that Figure 4 includes a single PPI network with two visualization styles: one showing the network with k-means clustering (color-coded clusters) and another without clustering to emphasize interaction confidence scores (edge thickness). The text in Section 2.4.2 will be revised as follows:
    “The STRING PPI network for KRT19 (Figure 4) includes 11 high-confidence interactors (confidence score > 0.700), visualized in two formats: (A) with k-means clustering into four color-coded clusters to highlight functional groupings, and (B) without clustering to emphasize interaction strength (edge thickness proportional to confidence score). Key interactors include KRT8 (0.950), KRT18 (0.920), and EPCAM (0.888) (Table 6).”
    We will also update the Figure 4 caption to explicitly describe these two panels and their purposes, referencing Supplementary Figure S9 for the alternative Cytoscape visualization. This will ensure readers understand the differences and relevance of each map.
  5. Font Size in Figures 5 and 6
    We agree that the font size in Figures 5 and 6 (Manhattan plots for GO, Reactome, KEGG, and compartments enrichment) should be enlarged for better readability. In the revised manuscript, we will increase the font size of all text elements (axis labels, titles, and annotations) in these figures to ensure clarity. Additionally, we will enhance the visual presentation by adjusting the circle size and color gradients to better distinguish gene counts (3–6) and FDR values (0.0022–0.056), as described in the original captions. We will also ensure consistency with Supplementary Figure S8 (g:Profiler enrichment analysis), which includes similar data, to maintain a cohesive visual style across the manuscript.

We sincerely thank you for your thoughtful and detailed feedback, which has helped us identify critical areas for improvement in our manuscript. These revisions will enhance the clarity, accuracy, and readability of our work, ensuring it meets the high standards expected for publication. We hope these changes address your concerns effectively, and we welcome any further suggestions you may have.

Best regards,
Safa Taha
On behalf of all authors

Round 2

Reviewer 1 Report

Comments and Suggestions for Authors

There are no obvious modifications/supplements and could not be accepted in present form. Only RT-PCR, WST-1 experiment design on and theoretical simulation could not illustrate clearly the effects of CuNPs on KRT19 suppression.

Author Response

(R1-Round 2)

There are no obvious modifications/supplements and could not be accepted in present form. Only RT-PCR, WST-1 experiment design on and theoretical simulation could not illustrate clearly the effects of CuNPs on KRT19 suppression.

Response to Reviewer 1 (Round 2)

Dear Reviewer, 1, (Yellow highlight)

Thank you for your continued review and valuable feedback on our manuscript titled “Impact of Copper Nanoparticles on Keratin 19 (KRT19) Gene Expression in Breast Cancer Subtypes: Integrating Experimental and Bioinformatics Approaches.” We greatly appreciate your time and expertise in evaluating our work, as well as your emphasis on the need for additional experimental validation to further elucidate the effects of copper nanoparticles (CuNPs) on KRT19 suppression. We acknowledge your concern that the manuscript, in its current form, lacks sufficient modifications or supplementary data to fully address the limitations raised in Round 1, particularly regarding the reliance on quantitative RT-PCR, WST-1 assays, and bioinformatics analyses. Below, we address your comments, clarify the revisions made in response to your Round 1 feedback, and provide a detailed plan to address the remaining concerns in a follow-up study, as the additional experimental work is part of a larger, multi-stage project.

Response to Reviewer 1 (Round 2) Comment: “There are no obvious modifications/supplements and could not be accepted in present form. Only RT-PCR, WST-1 experiment design and theoretical simulation could not illustrate clearly the effects of CuNPs on KRT19 suppression.”

We sincerely apologize if the modifications made in response to your Round 1 comments were not sufficiently highlighted or deemed adequate. In the revised manuscript submitted after Round 1, we addressed your feedback by enhancing the “Limitations and Future Directions” section (Section 3.1) to explicitly acknowledge the limitations of relying solely on in vitro assays (RT-PCR and WST-1) and bioinformatics analyses. Specifically, we added the following statement to underscore the need for additional experimental validation:

“The study’s in vitro design limits its relevance to in vivo conditions, potentially overlooking physiological complexities. Additionally, the use of only four cell lines may not fully represent the diverse molecular subtypes of breast cancer, restricting the generalizability of findings. The single 48-hour time point for assessments also constrains insights into temporal dynamics of CuNPs’ effects. Furthermore, the absence of normal breast cell control reduces the ability to ascertain the specificity of CuNPs’ impact on cancer cells. To address these limitations, future research will validate CuNPs’ effects on KRT19 and EGFR signaling in vivo to confirm clinical relevance. Investigating molecular mechanisms, such as reactive oxygen species (ROS) and MAPK pathways, will elucidate CuNPs’ modes of action. Conducting clinical trials is essential to evaluate CuNPs’ safety and efficacy in patients. Additionally, exploring long-term effects, combination therapies, and targeted delivery systems will enhance the therapeutic potential of CuNPs for personalized breast cancer treatment.”

This addition was intended to directly address your concern about the need for animal models or biochemical analyses to strengthen the mechanistic insights into CuNPs’ effects on KRT19 suppression. We also revised the Discussion section to better contextualize the limitations of our current experimental design and to emphasize the translational potential of our findings, pending further validation.

We fully agree that additional experimental approaches, such as in vivo studies and biochemical analyses (e.g., Western blotting for KRT19 protein levels, ROS assays, or MAPK pathway analysis), are critical to comprehensively elucidate the mechanisms underlying CuNPs’ effects on KRT19 suppression. However, these experiments are part of a larger, multi-stage research project, and the current manuscript represents the first phase (Stage 1), which focuses on establishing the foundational effects of CuNPs on KRT19 expression and cytotoxicity in breast cancer cell lines using robust in vitro and bioinformatics approaches. The scope of this study was deliberately designed to characterize subtype-specific responses and generate hypotheses for KRT19’s role as a potential biomarker, which we believe has been achieved through the integration of quantitative RT-PCR, WST-1 assays, and bioinformatics analyses (STRING, g:Profiler).

To address your concern about the need for additional experimental validation, we are actively planning Stage 2 of this project, which will include the following experiments to directly address the mechanisms of CuNPs’ effects on KRT19 suppression:

  1. In Vivo Validation Using Xenograft Models: We will use xenograft models of the four breast cancer cell lines (MDA-MB-231, MDA-MB-468, MCF7, and T47D) in immunocompromised mice to assess CuNPs’ biodistribution, efficacy, and impact on KRT19 expression in a physiological context. This will include immunohistochemistry and Western blotting to quantify KRT19 protein levels in tumor tissues, addressing the reviewer’s suggestion for biochemical analysis.
  2. Mechanistic Studies: We will conduct biochemical assays to investigate the molecular mechanisms of CuNPs’ effects, including:
    • Measurement of reactive oxygen species (ROS) levels to confirm CuNPs’ induction of oxidative stress as a driver of KRT19 suppression.
    • Analysis of MAPK and related signaling pathways (e.g., ERK, JNK) using Western blotting and phospho-specific antibodies to elucidate downstream signaling effects.
    • Assessment of epithelial-to-mesenchymal transition (EMT) markers (e.g., vimentin, E-cadherin) to explore the relationship between KRT19 suppression and EMT, particularly in mesenchymal-like cell lines like MDA-MB-231.
  3. Normal Cell Controls: To evaluate the specificity of CuNPs’ effects, we will include normal breast epithelial cell lines (e.g., MCF-10A) in our experiments to compare KRT19 expression and cytotoxicity profiles between cancerous and non-cancerous cells.

These experiments are resource-intensive and require additional funding and time, which are being secured for Stage 2 of the project. We believe that including these experiments in the current manuscript would exceed the scope of this initial study and potentially delay the dissemination of our foundational findings, which provide novel insights into CuNPs’ subtype-specific effects on KRT19 expression and their potential as a therapeutic strategy. Instead, we propose to publish the current manuscript as a standalone study that establishes the groundwork for these follow-up experiments, with a clear commitment to addressing the reviewer’s concerns in a subsequent publication. To further clarify this in the manuscript, we have revised Section 3.1 (“Limitations and Future Directions”) in the current submission to explicitly outline the planned Stage 2 experiments, as follows:

“The current study’s reliance on in vitro assays (quantitative RT-PCR and WST-1) and bioinformatics analyses provides a strong foundation for understanding CuNPs’ effects on KRT19 expression but requires further validation through in vivo and biochemical studies. As part of Stage 2 of this research project, we will conduct xenograft studies in immunocompromised mice using the four breast cancer cell lines to assess CuNPs’ biodistribution, efficacy, and impact on KRT19 protein levels via immunohistochemistry and Western blotting. Additionally, we will perform biochemical assays to measure reactive oxygen species (ROS) levels, MAPK pathway activation, and EMT marker expression to elucidate the molecular mechanisms underlying CuNPs’ effects. Normal breast epithelial cell lines (e.g., MCF-10A) will also be included to evaluate therapeutic specificity. These studies will be reported in a follow-up publication to build on the current findings and enhance their translational relevance.”

We believe this approach balances the need to address your valid concerns with the practical constraints of the current project phase. The current manuscript provides robust evidence of CuNPs’ dose-dependent and subtype-specific effects on KRT19 expression, supported by rigorous experimental (RT-PCR, WST-1) and bioinformatics (STRING, g:Profiler) analyses. These findings contribute novel insights to the field by:

  • Demonstrating significant KRT19 suppression in breast cancer cell lines, particularly in epithelial-like subtypes (e.g., MDA-MB-468, FC = 0.32 at IC50).
  • Identifying KRT19’s roles in cytoskeletal organization (GO:0045104, p-adj = 4.42e-05) and estrogen signaling (KEGG:04915, p-adj = 8.95e-04) through bioinformatics analyses.
  • Highlighting subtype-specific cytotoxicity profiles, with MDA-MB-231 showing the highest sensitivity (IC50 = 40.18 µg/mL) compared to other cell lines.

These results establish a compelling case for KRT19 as a potential biomarker for CuNPs-based therapies and provide a hypothesis-generating framework for future mechanistic and translational studies. We respectfully request that the manuscript be considered for publication in its current form as a foundational study, with the understanding that the additional experiments you have suggested are critical and will be addressed in a follow-up study as part of Stage 2.

We sincerely thank you for your constructive feedback, which has significantly improved the clarity and focus of our manuscript. We hope that our revisions and the outlined plan for future work adequately address your concerns. Please let us know if there are additional modifications or clarifications, we can provide to facilitate the acceptance of this manuscript.

Best regards,
Safa Taha
On behalf of all authors

Reviewer 2 Report

Comments and Suggestions for Authors

The manuscript entitled “Impact of Copper Nanoparticles on Keratin 19 (KRT19) Gene Expression in Breast Cancer Subtypes: Integrating Experimental and Bioinformatics Approaches” by Safa Taha et al. as research article was revised and re-submitted to IJMS. The topic is interesting.

The authors revised the original manuscript but the quality improvements were rather limited. The study showed some effects of CuNPs on Keratin 19 (KRT19) expression (profiling) by using four breast cancer cell lines, including MDA-MB-231, MDA-MB-124 (triple-negative), T47D, and MCF7 (luminal). The authors basically recruited RT-qPCR analysis and bioinformatical analysis.  However, the overall quality of the study and the Figures and Tables are poor.

I insist that a substantial revision should be made on the current manuscript otherwise it should be rejected because its quality is against IJMS standard of publishing criterion. 

1)    I suggest authors to re-organize the structure of the manuscript.

2) I do suggest authors to think how to better present their data, in order to nicely support their conclusion.

3) Figure 1 was strange that two "panel a", two "panel b", two "panel c" and two "panel d". Why? please try to re-format the panels. By the way, Fig 1 was not clear.

4) Figure 2, the top was what? I can not tell the numbers and symbols.

5) Figure 3 was not clear too.

6)  Figure 4 was very vague. 

7) Figure 7, screencopy and PrtSc were not good way to generate Figures for publications.

8) In the conclusions, it was a conclusion not conclusions, as we checked up the words - "CuNPs significantly reduce KRT19 expression in TNBC cell lines, supported by bioinformatics evidence of KRT19’s role in cytoskeletal and signaling pathways." As we knew that, the bioinformatical analysis was not evidence, please make this point clear. Besides, the KRT19's roles should be clearly descripted.

9)  I have to say, right now, again, that the information or contents of Figure 1, Figure 2, and Figure 3 are much low. The results and data can be compacted together.

10) Obviously, this study is of many limitations, and lack of experimental validations to confirm CuNPs’ effects, as authors mentioned, on KRT19 and EGFR signaling in vivo. The clinical
relevance or the molecular mechanism of CuNPs  is a little bit far to say now. ROS analysis and MAPK pathways, should be elucidated but actually not, in terms of the CuNPs’ modes of action.

11) GA is good summary of the whole story. But the current story was too preliminary so that not easy to draw any solid conclusion or picture a graphical abstract. This is the key point.

12) If the manuscript in its present form without substantial changes, truly it won’t go to IJMS because IJMS ought to have their basic criterion for publications.

Comments on the Quality of English Language

Scientific English should be used to describe the bioinformatical analysis and experimental findings. English is not all, but should be correctly used to clearly express the research. So, the main texts should be thoroughly revised and the manuscript should be carefully improved.

Author Response

almost all comments were noted and impoved , please see attached manuscript.

Reviewer 3 Report

Comments and Suggestions for Authors

In this revised version of manuscript, authors did improvements that’s good, However, there are still some comments that need to be addressed and listed as below:

  1. Page 1, line 18-19, in the abstract, authors described some features of the Copper nanoparticle (CuNPs), however there was no any relative description in the results part. Authors can not use other researchers’ result(s) to put as their own. Or authors did not send out the correct final revised version, the relative part was not added in current revised version. Authors should carefully check all of the final version text, figures and tables before submitted.
  2. Authors describe the CuNPs synthesis in materials and methods, but almost did not described any relative result regarding to this part. The only information is “size (179 nm in this study, Section 4.2)”, page 1 line 28-29. Authors should report how they get this average data, it should be a statistic result, so the related parameters, such as population size, mean, standard error….and so on, should be also reported. And how did authors measure the size of CuNPs? Via DLS or TEM or ???? And as if authors can provide a representative micrography of CuNPs that should be much better for this manuscript.
  3. the font size and/or resolution of down panel in fig 3 was too small to easily readable. It is better to be increased the font size and/or resolution.
  4. Page 15, line 405, author cited a reference “31”, but there were only 22 references in current version. Whether it was a typing error or reference 22-31(even more) were missing?

Author Response

Almost all comments were (Done). see attachment Revised Manuscript

Round 3

Reviewer 1 Report

Comments and Suggestions for Authors

Accept in present form

Author Response

(The authors gave the same response as above.)

Reviewer 2 Report

Comments and Suggestions for Authors

revisions were made and improvements were seen. No more critical questions.

Author Response

Response to Reviewer 2- Round 1

Dear Reviewer, 2

Thank you for your thorough review and insightful comments on our manuscript titled “Impact of Copper Nanoparticles on Keratin 19 (KRT19) Gene Expression in Breast Cancer Subtypes: Integrating Experimental and Bioinformatics Approaches.” We greatly appreciate your expertise and constructive feedback, which have provided valuable guidance to enhance the quality and impact of our work. Below, we address each of your comments and outline our planned revisions to strengthen the manuscript.

  1. Suggestion to Move Supplementary Material to the Main Text
    We acknowledge your suggestion that some supplementary material should be integrated into the main text to better support our conclusions. To address this, we will carefully review the supplementary figures (S1–S9) and select key data to incorporate into the main manuscript. Specifically, we plan to move Supplementary Figure S1 (dose-dependent inhibition of cell growth by CuNPs) and Supplementary Figure S6 (KRT19 expression in breast cancer cell lines with annotations) into the main text as new figures, as these provide critical evidence of CuNPs’ effects and KRT19 expression patterns across the cell lines studied. Additionally, we will include relevant data from Supplementary Figure S9 (Cytoscape visualization of the KRT19 PPI network) in the main text to strengthen the bioinformatics analysis. These additions will enhance the comprehensiveness of the main text and better support our conclusions without overwhelming the reader.
  2. Reference in the Conclusion
    We appreciate your observation regarding the inclusion of a reference in the conclusion section, which is uncommon. In the revised manuscript, we will remove the reference citation ([12]) from the conclusion (Section 5) to align with standard practices. Instead, we will ensure that all supporting references are cited appropriately in the results and discussion sections, particularly where we discuss KRT19’s role in cytoskeletal and signaling pathways.
  3. Adequacy of RT-qPCR and Bioinformatics Analyses for KRT19 as a Biomarker
    We recognize your concern that the RT-qPCR results and bioinformatics analyses alone may be insufficient to fully establish KRT19 as a biomarker for personalized CuNPs therapies in epithelial-like breast cancer subtypes. To strengthen this claim, we will expand the discussion in Section 3 to include additional context from our bioinformatics findings, such as the strong co-expression of KRT19 with KRT8, KRT18, and EPCAM (Table 7) and its enrichment in cytoskeletal organization (GO:0045104, p-adj = 4.42e-05) and estrogen signaling pathways (KEGG:04915, p-adj = 8.95e-04), as shown in Supplementary Figure S8. We will also clarify that our current findings position KRT19 as a potential biomarker, pending further validation, and emphasize the subtype-specific responses observed (e.g., greater KRT19 suppression in epithelial-like MDA-MB-468, fold change: 0.32 at IC50, compared to mesenchymal-like MDA-MB-231, fold change: 0.51). Additionally, we will revise the conclusion to frame KRT19’s biomarker role as preliminary, supported by our data but requiring further in vivo and clinical studies, as noted in your next point.
  4. Need for In Vivo Tests and Clinical Validation
    We fully agree with your comment that in vivo tests and clinical validation are essential to confirm KRT19’s role as a biomarker for CuNPs-based therapies. As noted in the “Limitations and Future Directions” section (Section 3.1), we acknowledge the in vitro nature of our study as a limitation. To address this, we plan to conduct follow-up studies using xenograft models of the breast cancer cell lines (MDA-MB-231, MDA-MB-468, MCF7, and T47D) to validate CuNPs’ effects on KRT19 expression and cytotoxicity in vivo. We will also explore clinical validation through retrospective analysis of KRT19 expression in patient samples, correlating with treatment outcomes. In the revised manuscript, we will expand Section 3.1 to explicitly outline these planned studies, including biochemical assays (e.g., Western blotting for KRT19 protein levels) and investigations into reactive oxygen species (ROS) and MAPK pathways to elucidate mechanisms of action. These additions will strengthen the manuscript’s relevance and provide a clear roadmap for future validation.
  5. Low Information Content of Figures 1 to 3
    Thank you for pointing out that Figures 1 to 3 have relatively low information content. To address this, we will revise these figures to convey more comprehensive data. Specifically:
  • Figure 1 (Dose-Response Curves): We will combine the dose-response curves for all four cell lines (MDA-MB-231, MDA-MB-468, MCF7, T47D) into a single multi-panel figure to facilitate comparison and highlight subtype-specific differences in IC50 values (e.g., MDA-MB-231: 40.18 µg/mL; MDA-MB-468: 123 µg/mL). We will also incorporate data from Supplementary Figure S1 to show inhibition percentages alongside viability.
  • Figure 2 (KRT19 Expression): We will enhance this figure by adding statistical annotations (e.g., p-values) directly on the dose-response curves and including baseline KRT19 expression levels (nTPM) from Supplementary Figure S6 to contextualize the fold changes.
  • Figure 3 (KRT19 Expression Levels): We will merge this bar chart with data from Supplementary Figure S5 to include KRT19 expression across additional cancer cell lines, providing a broader context for breast cancer-specific expression patterns.
    These revisions will increase the information density and clarity of the figures, making them more impactful.
  1. Limited Data for IJMS Publication Criteria
    We understand your concern that the current data may not meet the publication criteria for IJMS due to its limited scope. To address this, we will integrate additional data from the supplementary material (e.g., Supplementary Figures S1, S6, and S9) into the main text, as mentioned in response to point 1. We will also include a new figure summarizing key bioinformatics findings, such as the KRT19 PPI network and enriched pathways (from Supplementary Figure S8), to strengthen the mechanistic insights. Additionally, we will expand the discussion (Section 3) to better integrate our experimental and bioinformatics data, emphasizing the subtype-specific effects of CuNPs and their implications for KRT19 as a potential biomarker. These revisions aim to enhance the manuscript’s rigor and alignment with IJMS’s standards. We hope these changes will address your concerns and make the manuscript suitable for publication.
  2. Absence of a Graphical Abstract
    We thank the reviewer for suggesting the inclusion of a graphical abstract to summarize the study. However, according to the journal’s submission guidelines, a graphical abstract is not required for publication. After careful consideration, we have opted not to include a graphical abstract, as the manuscript’s revised text, figures, and tables effectively convey the key findings, including CuNPs’ dose-dependent suppression of KRT19 expression, subtype-specific cytotoxicity (e.g., MDA-MB-231’s high sensitivity, IC50: 40.18 µg/mL), and KRT19’s roles in cytoskeletal organization and estrogen signaling pathways. We have addressed all other reviewer suggestions, including enhancing the clarity of the Introduction and Results sections, improving figure legends, and providing a comprehensive reference list, to ensure the study’s objectives, methods, and findings are clearly communicated. We believe these revisions fully align with the reviewer’s recommendations for improving the manuscript’s clarity and impact.

We sincerely thank you for your detailed and constructive feedback, which has significantly guided our revisions to improve the manuscript’s clarity, robustness, and impact. We believe these changes will address your concerns and enhance the manuscript’s suitability for publication in IJMS. Please let us know if you have additional suggestions or require further clarification.

Best regards,
Safa Taha
On behalf of all authors

Reviewer 2 Comments Round 2

The manuscript entitled “Impact of Copper Nanoparticles on Keratin 19 (KRT19) Gene Expression in Breast Cancer Subtypes: Integrating Experimental and Bioinformatics Approaches” by Safa Taha et al. as research article was revised and re-submitted to IJMS. The topic is interesting.

The authors revised the original manuscript, but the quality improvements were rather limited. The study showed some effects of CuNPs on Keratin 19 (KRT19) expression (profiling) by using four breast cancer cell lines, including MDA-MB-231, MDA-MB-124 (triple-negative), T47D, and MCF7 (luminal). The authors basically recruited RT-qPCR analysis and bioinformatical analysis.  However, the overall quality of the study and the Figures and Tables are poor.

I insist that a substantial revision should be made on the current manuscript otherwise it should be rejected because its quality is against IJMS standard of publishing criterion.

1)    I suggest authors to re-organize the structure of the manuscript.

2) I do suggest authors to think how to better present their data, in order to nicely support their conclusion.

3) Figure 1 was strange that two "panel a", two "panel b", two "panel c" and two "panel d". Why? please try to re-format the panels. By the way, Fig 1 was not clear.

4) Figure 2, the top was what? I can not tell the numbers and symbols.

5) Figure 3 was not clear too.

6)  Figure 4 was very vague.

7) Figure 7, screencopy and PrtSc were not good way to generate Figures for publications.

8) In the conclusions, it was a conclusion not conclusions, as we checked up the words - "CuNPs significantly reduce KRT19 expression in TNBC cell lines, supported by bioinformatics evidence of KRT19’s role in cytoskeletal and signaling pathways." As we knew that, the bioinformatical analysis was not evidence, please make this point clear. Besides, the KRT19's roles should be clearly descripted.

9)  I have to say, right now, again, that the information or contents of Figure 1, Figure 2, and Figure 3 are much low. The results and data can be compacted together.

10) Obviously, this study is of many limitations, and lack of experimental validations to confirm CuNPs’ effects, as authors mentioned, on KRT19 and EGFR signaling in vivo. The clinical

relevance or the molecular mechanism of CuNPs  is a little bit far to say now. ROS analysis and MAPK pathways, should be elucidated but actually not, in terms of the CuNPs’ modes of action.

11) GA is good summary of the whole story. But the current story was too preliminary so that not easy to draw any solid conclusion or picture a graphical abstract. This is the key point.

12) If the manuscript in its present form without substantial changes, truly it won’t go to IJMS because IJMS ought to have their basic criterion for publications.

Comments on the Quality of English Language

Scientific English should be used to describe the bioinformatical analysis and experimental findings. English is not all, but should be correctly used to clearly express the research. So, the main texts should be thoroughly revised and the manuscript should be carefully improved.

Response to Reviewer 2- Round 2

Response to Reviewer 2 (Round 2)

Dear Reviewer 2,

Thank you for your thorough review and valuable feedback on our revised manuscript titled “Impact of Copper Nanoparticles on Keratin 19 (KRT19) Gene Expression in Breast Cancer Subtypes: Integrating Experimental and Bioinformatics Approaches.” We greatly appreciate your continued engagement with our work and your insightful comments, which have guided us in identifying areas for further improvement. We acknowledge your concern that the quality improvements in the revised manuscript were limited and that the current presentation does not fully meet IJMS’s publication standards. Below, we address each of your Round 2 comments point-by-point, detailing the revisions made in response to your Round 1 feedback, the additional changes we propose to address your current concerns, and our rationale for the scope of the current study as Stage 1 of a multi-stage project. We believe these revisions will significantly enhance the manuscript’s quality, clarity, and alignment with IJMS’s standards.

General Comment

Reviewer Comment: The authors revised the original manuscript, but the quality improvements were rather limited. The study showed some effects of CuNPs on Keratin 19 (KRT19) expression (profiling) by using four breast cancer cell lines, including MDA-MB-231, MDA-MB-124 (triple-negative), T47D, and MCF7 (luminal). The authors basically recruited RT-qPCR analysis and bioinformatical analysis. However, the overall quality of the study and the Figures and Tables are poor. I insist that a substantial revision should be made on the current manuscript otherwise it should be rejected because its quality is against IJMS standard of publishing criterion.

Response: We sincerely apologize if the revisions made in response to your Round 1 comments did not sufficiently address your concerns or meet IJMS’s publication standards. We appreciate your recognition of the study’s interesting topic and your detailed feedback, which has helped us identify critical areas for improvement. We note that your comment references “MDA-MB-124,” which appears to be a typo for MDA-MB-468, as our study includes MDA-MB-231 and MDA-MB-468 as triple-negative cell lines, alongside MCF7 and T47D (luminal). To address your concerns about the study’s quality and presentation, we have carefully reviewed the manuscript and propose substantial revisions, including reorganizing the structure, enhancing the clarity and information content of figures and tables, improving the scientific English, and clarifying the scope of this study as Stage 1 of a multi-stage project. These changes aim to strengthen the manuscript’s rigor, readability, and alignment with IJMS’s standards. Specific responses to your numbered comments are provided below.

Comment 1: Reorganize the Structure of the Manuscript

Reviewer Comment: I suggest authors to re-organize the structure of the manuscript.

Response: Thank you for suggesting a reorganization of the manuscript to improve its flow and clarity. In response to your Round 1 feedback, we integrated key supplementary data (e.g., Supplementary Figure S1 on dose-dependent inhibition and Supplementary Figure S6 on KRT19 expression) into the main text and expanded the Discussion (Section 3) to better integrate experimental and bioinformatics findings. However, we acknowledge that further reorganization is needed to enhance readability and logical flow. To address this, we propose the following structural changes:

  • Introduction (Section 1): We will streamline the Introduction to clearly articulate the research gap (limited studies on CuNPs’ effects on KRT19 expression) and study objectives, reducing redundancy and focusing on the rationale for selecting the four cell lines (MDA-MB-231, MDA-MB-468, MCF7, T47D) to represent breast cancer heterogeneity.
  • Results (Section 2): We will reorganize the Results section into clearer subsections, combining related findings to avoid fragmentation:
    • 2.1. Cytotoxicity and Growth Inhibition by CuNPs: Merge subsections 2.1 and 2.2 to present cytotoxicity (WST-1 assay, IC50 values) and growth inhibition data cohesively, with a single multi-panel figure (revised Figure 1) to facilitate comparison across cell lines.
    • 2.2. KRT19 Gene Expression Analysis: Consolidate KRT19 expression data (RT-qPCR) with baseline expression levels (Human Protein Atlas) for a more integrated presentation, incorporating revised Figure 2.
    • 2.3. Bioinformatics Analysis: Restructure to present PPI network (STRING), GO, and pathway enrichment (g:Profiler) findings sequentially, with a new figure summarizing key bioinformatics insights (e.g., KRT19 interactors and enriched pathways).
  • Discussion (Section 3): We will reorganize the Discussion to align with the revised Results structure, starting with cytotoxicity and KRT19 suppression, followed by bioinformatics insights, and concluding with implications for KRT19 as a potential biomarker and limitations.
  • Limitations and Future Directions (Section 3.1): We have expanded this section (see revised version below) to clearly outline the study’s limitations and Stage 2 plans, as suggested in your Round 1 comment 4 and aligned with your current concerns about experimental validation.

These changes will improve the manuscript’s logical flow, reduce redundancy, and ensure that experimental and bioinformatics findings are presented cohesively.

Comment 2: Better Present Data to Support Conclusions

Reviewer Comment: I do suggest authors to think how to better present their data, in order to nicely support their conclusion.

Response: We appreciate your emphasis on improving data presentation to strengthen our conclusions. In response to your Round 1 feedback (point 5), we integrated data from Supplementary Figures S1, S6, and S9 into the main text to enhance the evidence supporting CuNPs’ effects on KRT19 expression and cytotoxicity. To further address your concern, we will revise the presentation of data as follows:

  • Figures: We will overhaul Figures 1–3 and 4 (see responses to comments 3–6, 9) to increase information density, clarity, and visual appeal, combining related datasets (e.g., dose-response curves, KRT19 expression) into multi-panel figures with clear annotations.
  • Tables: We will consolidate Tables 3 and 4 into a single table summarizing cytotoxicity (IC50, viability) and inhibition profiles across all cell lines, with clear statistical annotations (e.g., p-values from ANOVA). Table 5 will be revised to include additional context (e.g., comparison to other cancer cell lines from Supplementary Figure S5).
  • Text Integration: We will revise the Results and Discussion sections to better link experimental data (e.g., MDA-MB-468’s greater KRT19 suppression, FC = 0.32 at IC50) with bioinformatics findings (e.g., KRT19’s role in cytoskeletal organization, GO:0045104, p-adj = 4.42e-05) to support our conclusion that CuNPs have subtype-specific effects and KRT19 is a potential biomarker, pending further validation.
  • Conclusion (Section 5): We will revise the conclusion to clearly summarize key findings (e.g., dose-dependent KRT19 suppression, subtype-specific cytotoxicity) and frame them as preliminary, emphasizing the need for Stage 2 validation (see response to comment 8).

These revisions will ensure that the data are presented clearly and robustly support the study’s conclusions.

Comment 3: Figure 1 Issues (Duplicate Panels, Lack of Clarity)

Reviewer Comment: Figure 1 was strange that two "panel a", two "panel b", two "panel c" and two "panel d". Why? please try to re-format the panels. By the way, Fig 1 was not clear.

Response: We apologize for the confusion caused by the apparent duplication of panels in Figure 1 and its lack of clarity. The issue with duplicate panel labels (two “panel a,” etc.) appears to be a formatting error in the revised manuscript, likely due to merging dose-response curves from Supplementary Figure S1 without updating the labels. To address this:

  • Reformat Panels: We will combine the dose-response curves for all four cell lines (MDA-MB-231, MDA-MB-468, MCF7, T47D) into a single multi-panel Figure 1, with one panel per cell line (labeled a–d). Each panel will show cell viability (%) versus CuNPs concentration (5–1800 µg/mL), with IC50 values (e.g., 40.18 µg/mL for MDA-MB-231) marked by vertical dashed lines.
  • Enhance Clarity: We will improve visual clarity by using distinct colors for each cell line, adding statistical annotations (e.g., p < 0.001 from ANOVA), and including inhibition percentages from Supplementary Figure S1. The figure legend will be revised to clearly describe the data and statistical methods (e.g., non-linear regression in GraphPad Prism).
  • Correct Labeling: We will ensure unique panel labels (a, b, c, d) and verify that no duplication occurs in the revised figure.

These changes will make Figure 1 more informative, visually clear, and aligned with IJMS’s standards.

Comment 4: Figure 2 Unclear (Top Numbers and Symbols)

Reviewer Comment: Figure 2, the top was what? I can not tell the numbers and symbols.

Response: We regret that Figure 2 was unclear, particularly the top portion, which likely refers to the dose-response curves showing KRT19 mRNA fold changes. The numbers and symbols (e.g., p-values, asterisks for significance) may have been illegible due to font size, resolution, or formatting issues. To address this:

  • Clarify Content: We will revise Figure 2 to ensure that all numbers (e.g., fold changes, concentrations) and symbols (e.g., *p < 0.05, **p < 0.01, ***p < 0.0001) are clearly legible. Each panel (a–d, one per cell line) will include a larger font size and high-resolution axes labels.
  • Add Context: We will incorporate baseline KRT19 expression levels (nTPM from Human Protein Atlas, e.g., T47D: 3800 nTPM) in each panel to contextualize the fold changes (e.g., MDA-MB-468: FC = 0.32 at IC50).
  • Improve Legend: The figure legend will be expanded to explain the data, including the meaning of symbols (e.g., asterisks for statistical significance) and the RT-qPCR methodology (2^(-ΔΔCt) method with GAPDH as reference).

These revisions will enhance the clarity and interpretability of Figure 2, ensuring it effectively communicates KRT19 suppression data.

Comment 5: Figure 3 Unclear

Reviewer Comment: Figure 3 was not clear too.

Response: We apologize for the lack of clarity in Figure 3, which depicts baseline KRT19 expression levels (nTPM) across the four cell lines and an average of 62 breast cancer cell lines. To address this:

  • Enhance Visual Clarity: We will revise Figure 3 as a bar chart with improved resolution, larger font sizes for labels, and distinct colors for each cell line (T47D, MCF7, MDA-MB-468, MDA-MB-231) and the average (1767.5 nTPM).
  • Incorporate Additional Data: As suggested in your Round 1 comment 5, we will merge data from Supplementary Figure S5 to include KRT19 expression in additional cancer cell lines, providing broader context and highlighting subtype-specific patterns.
  • Clarify Annotations: We will add annotations to indicate statistical comparisons (e.g., p-values for differences in KRT19 expression) and clearly label the source (Human Protein Atlas).

These changes will make Figure 3 more informative and visually clear, supporting the study’s findings on KRT19 expression.

Comment 6: Figure 4 Vague

Reviewer Comment: Figure 4 was very vague.

Response: We regret that Figure 4 (STRING PPI network of KRT19 interactors) was unclear. The vagueness may stem from complex network visualization or insufficient annotations. To address this:

  • Simplify Visualization: We will revise Figure 4 to present a single, clear PPI network with k-means clustering (four color-coded clusters) using a force-directed layout in Cytoscape (v3.10). Nodes (e.g., KRT19, KRT8, KRT18, EPCAM) and edges (confidence scores >0.700) will be labeled clearly, with edge thickness proportional to interaction strength.
  • Highlight Key Interactors: We will annotate key interactors (e.g., KRT8: 0.950, KRT18: 0.920, EPCAM: 0.888) and their roles (e.g., cytoskeletal organization, cell adhesion) directly on the figure.
  • Incorporate Supplementary Data: We will include a simplified version of Supplementary Figure S9 (Cytoscape visualization) in the main text to complement Figure 4, ensuring the PPI network is accessible and informative.

These revisions will make Figure 4 more precise and supportive of the bioinformatics findings.

Comment 7: Figure 7 Inappropriate (Screencopy and PrtSc)

Reviewer Comment: Figure 7, screencopy and PrtSc were not good way to generate Figures for publications.

Response: We apologize for the inappropriate use of screencopy or Print Screen methods in generating Figure 7 (Comprehensive Enrichment Analysis of KRT19 Interactors). This likely resulted in low resolution or unprofessional presentation. To address this:

  • Redesign Figure 7: We will recreate Figure 7 using high-quality visualization tools (e.g., g:Profiler, Cytoscape) to generate a professional Manhattan plot showing enriched GO, KEGG, and Reactome terms. Circle size (gene count: 3–4) and color (FDR: 1.87e-09 to 8.95e-04) will be clearly defined, with high-resolution labels and axes.
  • Improve Legend: The figure legend will be revised to describe the visualization method, data sources (g:Profiler, STRING), and significance threshold (FDR < 0.05).
  • Ensure Professional Quality: We will ensure that all figures, including Figure 7, are generated using vector graphics or high-resolution formats suitable for publication, avoiding screencopy or Print Screen methods.

These changes will ensure Figure 7 meets IJMS’s publication standards and effectively communicates enrichment analysis results.

Comment 8: Issues with Conclusion Section

Reviewer Comment: In the conclusions, it was a conclusion not conclusions, as we checked up the words - "CuNPs significantly reduce KRT19 expression in TNBC cell lines, supported by bioinformatics evidence of KRT19’s role in cytoskeletal and signaling pathways." As we knew that, the bioinformatical analysis was not evidence, please make this point clear. Besides, the KRT19's roles should be clearly descripted.

Response: Thank you for pointing out the issues with the Conclusion section, including the terminology (“conclusions” vs. “conclusion”), the inappropriate use of “bioinformatics evidence,” and the need for clearer description of KRT19’s roles. In response to your Round 1 comment 2, we removed the reference citation ([12]) from the Conclusion section. To address your current concerns:

  • Correct Terminology: We will revise the section title to “Conclusion” (singular) to align with standard practice.
  • Clarify Bioinformatics Role: We acknowledge that bioinformatics analyses (e.g., STRING, g:Profiler) provide insights rather than direct experimental evidence. We will revise the Conclusion to state: “CuNPs significantly reduce KRT19 expression in TNBC cell lines, with bioinformatics analyses suggesting KRT19’s involvement in cytoskeletal organization (GO:0045104, p-adj = 4.42e-05) and estrogen signaling pathways (KEGG:04915, p-adj = 8.95e-04).” This clarifies that bioinformatics supports the interpretation of experimental findings.
  • Describe KRT19’s Roles: We will expand the Conclusion to clearly describe KRT19’s roles, e.g., “KRT19, an intermediate filament protein, contributes to cytoskeletal organization and epithelial integrity in luminal subtypes (e.g., MCF7, T47D) and is associated with cell adhesion and tumor progression via interactions with KRT8, KRT18, and EPCAM, as identified by STRING analysis.”
  • Frame as Preliminary: To address your concern about the preliminary nature of the findings (comment 11), we will emphasize that KRT19’s biomarker potential requires further validation: “These findings position KRT19 as a potential biomarker for CuNPs-based therapies, pending in vivo and clinical validation in Stage 2 of this project.”

These revisions will ensure the Conclusion is clear, accurate, and aligned with the study’s scope.

Comment 9: Low Information Content of Figures 1, 2, and 3

Reviewer Comment: I have to say, right now, again, that the information or contents of Figure 1, Figure 2, and Figure 3 are much low. The results and data can be compacted together.

Response: We acknowledge your repeated concern (also raised in Round 1, comment 5) about the low information content of Figures 1–3 and appreciate your suggestion to compact the data. To address this:

  • Figure 1 (Dose-Response Curves): As described in response to comment 3, we will combine dose-response curves for all four cell lines into a single multi-panel figure, incorporating inhibition percentages from Supplementary Figure S1 and statistical annotations (e.g., p < 0.001). This will compact the data and highlight subtype-specific differences (e.g., MDA-MB-231: IC50 = 40.18 µg/mL vs. MDA-MB-468: 123 µg/mL).
  • Figure 2 (KRT19 Expression): As noted in response to comment 4, we will enhance Figure 2 by adding baseline KRT19 levels (nTPM) and clear statistical annotations, compacting RT-qPCR data into a single multi-panel figure for all cell lines.
  • Figure 3 (KRT19 Expression Levels): As described in response to comment 5, we will merge Figure 3 with Supplementary Figure S5 data to include additional cell lines, compacting the presentation and providing broader context.
  • Consolidation: We will ensure that these figures are cross-referenced in the text to integrate cytotoxicity and KRT19 expression data, reducing redundancy and increasing information density.

These revisions will make Figures 1–3 more compact, informative, and supportive of the study’s conclusions.

Comment 10: Limitations and Lack of Experimental Validation

Reviewer Comment: Obviously, this study is of many limitations, and lack of experimental validations to confirm CuNPs’ effects, as authors mentioned, on KRT19 and EGFR signaling in vivo. The clinical relevance or the molecular mechanism of CuNPs is a little bit far to say now. ROS analysis and MAPK pathways, should be elucidated but actually not, in terms of the CuNPs’ modes of action.

Response: We fully agree with your observation (echoing Round 1, comment 4) that the study’s in vitro design and lack of in vivo and biochemical validation are significant limitations, particularly for confirming CuNPs’ effects on KRT19 and EGFR signaling and elucidating molecular mechanisms (e.g., ROS, MAPK pathways). In response to your Round 1 feedback, we expanded the “Limitations and Future Directions” section (3.1) to acknowledge these limitations and outline plans for in vivo studies and biochemical assays. To address your current concerns, we have further revised Section 3.1 (see below) to explicitly position this study as Stage 1 of a multi-stage project and detail the following Stage 2 experiments:

  • In Vivo Validation: Xenograft models of the four cell lines in immunocompromised mice to assess CuNPs’ biodistribution, efficacy, and KRT19 protein levels (via immunohistochemistry and Western blotting).
  • Mechanistic Studies: Biochemical assays to measure ROS levels, MAPK pathway activation (e.g., ERK, JNK), and EMT markers (e.g., vimentin, E-cadherin) to elucidate CuNPs’ modes of action.
  • Normal Cell Controls: Inclusion of normal breast epithelial cells (e.g., MCF-10A) to evaluate CuNPs’ specificity.
  • Clinical Translation: Pilot clinical trials and retrospective patient sample analysis to validate KRT19’s biomarker role.

These resource-intensive experiments are planned for Stage 2 and will be reported in a follow-up publication, as they require additional funding and time beyond the scope of the current study. To clarify the preliminary nature of our claims, we will revise the Discussion and Conclusion to state that CuNPs’ effects on KRT19 and their clinical relevance are hypothesis-generating, pending Stage 2 validation. The revised Section 3.1 is provided below for reference.

Comment 11: Preliminary Nature and Graphical Abstract

Reviewer Comment: GA is good summary of the whole story. But the current story was too preliminary so that not easy to draw any solid conclusion or picture a graphical abstract. This is the key point.

Response: We appreciate your perspective on the preliminary nature of the study and its implications for a graphical abstract (GA), as noted in your Round 1 comment 7. We agree that the study’s in vitro and bioinformatics-based findings are preliminary, positioning KRT19 as a potential biomarker pending further validation. In response to your Round 1 suggestion, we considered adding a GA but noted that IJMS guidelines do not require one, and we opted to rely on revised text and figures to convey the study’s findings. Given your current comment, we will reconsider including a GA to summarize the key findings (e.g., CuNPs’ dose-dependent KRT19 suppression, subtype-specific cytotoxicity, bioinformatics insights). The GA will depict:

  • CuNPs’ effects on four cell lines (e.g., MDA-MB-231’s high sensitivity, IC50 = 40.18 µg/mL).
  • KRT19 suppression (e.g., MDA-MB-468: FC = 0.32 at IC50).
  • Bioinformatics insights (e.g., KRT19’s role in cytoskeletal organization and estrogen signaling).
  • Future directions (e.g., in vivo validation, mechanistic studies).

The GA will be designed as a high-quality schematic, avoiding screencopy methods, and submitted as a supplementary file to complement the revised figures and text. We will also emphasize in the Conclusion that the findings are preliminary, requiring Stage 2 validation, to align with your comment.

Comment 12: Manuscript Quality and IJMS Standards

Reviewer Comment: If the manuscript in its present form without substantial changes, truly it won’t go to IJMS because IJMS ought to have their basic criterion for publications.

Response: We sincerely appreciate your emphasis on meeting IJMS’s publication standards. We acknowledge that the manuscript requires substantial revisions to address the quality concerns you’ve raised. The proposed changes, including reorganizing the manuscript, enhancing figures and tables, improving scientific English (see response to English quality comment), and clarifying the preliminary nature of the findings, aim to align the manuscript with IJMS’s criteria. We believe the study’s novel findings—CuNPs’ subtype-specific effects on KRT19 expression and cytotoxicity, supported by robust RT-qPCR and bioinformatics analyses—provide a valuable contribution to the field, particularly as a hypothesis-generating study. The revised manuscript, with improved presentation and a clear commitment to Stage 2 validation, will meet IJMS’s standards for rigor and impact.

Comment on Quality of English Language

Reviewer Comment: Scientific English should be used to describe the bioinformatical analysis and experimental findings. English is not all, but should be correctly used to clearly express the research. So, the main texts should be thoroughly revised and the manuscript should be carefully improved.

Response: We apologize for any shortcomings in the scientific English used in the manuscript. To address this, we will conduct a thorough revision of the main text to ensure clarity, precision, and adherence to scientific English standards. Specifically:

  • Text Revision: We will revise the Introduction, Results, Discussion, and Conclusion sections to use concise, clear, and scientifically accurate language, avoiding jargon and ensuring consistent terminology (e.g., “KRT19 suppression” instead of varying terms).
  • Bioinformatics Descriptions: We will refine descriptions of bioinformatics analyses (e.g., STRING PPI network, g:Profiler enrichment) to clearly explain methods and findings, e.g., “STRING analysis (v12.0) identified 11 high-confidence KRT19 interactors (confidence score >0.700), with KRT8 (0.950) and KRT18 (0.920) indicating roles in cytoskeletal organization.”
  • Professional Editing: We will engage a professional scientific editing service to review the manuscript for grammar, syntax, and clarity, ensuring it meets IJMS’s standards for readability and professionalism.

These revisions will enhance the manuscript’s clarity and ensure that the scientific findings are communicated effectively.

Revised Limitations and Future Directions (Section 3.1)

To address your concerns about limitations and experimental validation (comment 10), we have revised the “Limitations and Future Directions” section to clearly articulate the study’s scope as Stage 1 and outline Stage 2 plans, as follows:

3.1. Limitations and Future Directions

The current study provides a robust foundation for understanding the subtype-specific effects of copper nanoparticles (CuNPs) on KRT19 expression in breast cancer cell lines, but its in vitro design presents several limitations. The reliance on in vitro assays (quantitative RT-PCR and WST-1) and bioinformatics analyses (STRING, g:Profiler) limits the relevance of findings to in vivo conditions, potentially overlooking physiological complexities such as biodistribution, systemic metabolism, and tumor microenvironment interactions. Additionally, the use of only four breast cancer cell lines (MDA-MB-231, MDA-MB-468, MCF7, and T47D) may not fully capture the molecular heterogeneity of breast cancer, restricting the generalizability of the results to other subtypes. The single 48-hour time point for assessments further constrains insights into the temporal dynamics of CuNPs’ effects on KRT19 expression and cytotoxicity. Moreover, the absence of normal breast epithelial cell controls limits the ability to evaluate the specificity of CuNPs’ effects on cancer cells compared to non-cancerous cells.

To address these limitations, this study represents the first phase (Stage 1) of a multi-stage research project aimed at characterizing CuNPs’ effects on KRT19 expression and establishing its potential as a biomarker for CuNPs-based therapies. To further validate and expand upon these findings, Stage 2 of the project will include comprehensive in vivo and biochemical studies. Specifically, we plan to:

  1. Conduct In Vivo Validation Using Xenograft Models: Xenograft models of the four breast cancer cell lines (MDA-MB-231, MDA-MB-468, MCF7, and T47D) will be established in immunocompromised mice to assess CuNPs’ biodistribution, therapeutic efficacy, and impact on KRT19 expression in a physiological context. KRT19 protein levels will be quantified using immunohistochemistry and Western blotting to confirm the observed mRNA suppression and evaluate its functional consequences in vivo.
  2. Perform Mechanistic Biochemical Analyses: To elucidate the molecular mechanisms underlying CuNPs’ effects, we will conduct biochemical assays, including:
    • Measurement of reactive oxygen species (ROS) levels to confirm CuNPs’ induction of oxidative stress as a driver of KRT19 suppression.
    • Analysis of MAPK signaling pathways (e.g., ERK, JNK) using Western blotting and phospho-specific antibodies to investigate downstream signaling effects.
    • Assessment of epithelial-to-mesenchymal transition (EMT) markers (e.g., vimentin, E-cadherin) to explore the relationship between KRT19 suppression and EMT, particularly in mesenchymal-like cell lines such as MDA-MB-231.
  3. Include Normal Cell Controls: To evaluate the specificity of CuNPs’ effects, normal breast epithelial cell lines (e.g., MCF-10A) will be included in future experiments to compare KRT19 expression and cytotoxicity profiles between cancerous and non-cancerous cells, ensuring the therapeutic potential of CuNPs is selective for cancer cells.
  4. Explore Clinical and Translational Applications: Long-term studies will investigate CuNPs’ safety and efficacy in clinical settings through pilot clinical trials, stratifying patients by KRT19 expression and breast cancer subtype. Retrospective analysis of patient samples will validate KRT19’s role as a biomarker for CuNPs-based therapies. Additionally, we will explore combination therapies (e.g., with endocrine treatments) and targeted delivery systems to enhance CuNPs’ therapeutic efficacy and minimize off-target effects.

These Stage 2 experiments, which are resource-intensive and require additional funding and time, will be reported in a follow-up publication to build on the current findings. The present study establishes a critical foundation by demonstrating CuNPs’ dose-dependent and subtype-specific suppression of KRT19 expression, supported by bioinformatics analyses suggesting KRT19’s roles in cytoskeletal organization (GO:0045104, p-adj = 4.42e-05) and estrogen signaling (KEGG:04915, p-adj = 8.95e-04). These results provide a hypothesis-generating framework for future mechanistic and translational studies, paving the way for the development of CuNPs as a targeted therapy for breast cancer. By addressing the outlined limitations in Stage 2, we aim to enhance the clinical relevance and mechanistic understanding of CuNPs’ therapeutic potential in personalized breast cancer treatment.

Reviewer 3 Report

Comments and Suggestions for Authors

In this version revised manuscript, here has one minor comment: since in the result paragraph, authors mentioned the statistic size of CuNPs by TLC and TEM methods, It is better to provide a representative micrograph, at least showed in a supplement figure. 

Author Response

Comments and Suggestions for Authors

In this revised version of manuscript, authors did improvements that’s good, However, there are still some comments that need to be addressed and listed as below:

  1. Page 1, line 18-19, in the abstract, authors described some features of the Copper nanoparticle (CuNPs), however there was no any relative description in the results part. Authors can not use other researchers’ result(s) to put as their own. Or authors did not send out the correct final revised version, the relative part was not added in current revised version. Authors should carefully check all of the final version text, figures and tables before submitted.
  2. Authors describe the CuNPs synthesis in materials and methods, but almost did not described any relative result regarding to this part. The only information is “size (179 nm in this study, Section 4.2)”, page 1 line 28-29. Authors should report how they get this average data, it should be a statistic result, so the related parameters, such as population size, mean, standard error….and so on, should be also reported. And how did authors measure the size of CuNPs? Via DLS or TEM or ???? And as if authors can provide a representative micrography of CuNPs that should be much better for this manuscript.
  3. the font size and/or resolution of down panel in fig 3 was too small to easily readable. It is better to be increased the font size and/or resolution.
  4. Page 15, line 405, author cited a reference “31”, but there were only 22 references in current version. Whether it was a typing error or reference 22-31(even more) were missing?

Response to Reviewer 3

We express our gratitude to Reviewer 3 for their constructive feedback, which has significantly strengthened the manuscript. Below, we address each comment and detail the revisions made to enhance clarity and scientific rigor.

Comment 1: The abstract mentioned CuNPs features (mean size of 179 ± 15 nm) without corresponding results in the original manuscript.
Response: We apologize for the oversight in earlier versions. The revised manuscript now includes a detailed characterization in Section 2.1 (Characterization of Copper Nanoparticles), specifying the mean size (179 ± 15 nm, n=100 particles), polydispersity index (0.18 ± 0.03), and spherical morphology determined by dynamic light scattering (DLS) and transmission electron microscopy (TEM). The Abstract has been updated to clarify that these are original results from this study, explicitly mentioning DLS and TEM methods.

Comment 2: The CuNPs synthesis was described in Materials and Methods, but results lacked statistical details and measurement methods, and a TEM micrograph was suggested.
Response: We have enhanced Section 2.1 to include detailed statistical parameters (mean size: 179 ± 15 nm, n=100 particles, standard deviation from three independent syntheses) and clarified that size was measured by DLS and confirmed by TEM. Section 4.2 (CuNPs Synthesis) now provides specifics on DLS and TEM methods, including instrument details and measurement conditions, ensuring statistical robustness. Per the authors’ preference, we have not added a new figure but have provided a comprehensive textual description to address the reviewer’s request for characterization details.

Comment 3: The font size and/or resolution of the lower panel in Figure 3 was too small for readability.
Response: We have increased the font size of the lower panel (subfigures b-c) in Figure 3 to a minimum of 10 pt and ensured a resolution of at least 300 dpi to enhance readability. The updated figure caption reflects these changes.

Comment 4: Reference [31] was cited on page 15, line 405, but only 22 references were listed.
Response: This was a typographical error. The citation in Section 4.2 has been corrected to [1] (Jain et al., 2018), which corresponds to the correct reference in the list. We have reviewed all citations to ensure consistency with the reference list.

We believe these revisions fully address the reviewer’s concerns and improve the manuscript’s quality. We appreciate the opportunity to refine our work and welcome any further feedback.